# *TUBA1A* tubulinopathy mutants disrupt neuron morphogenesis and override XMAP215/Stu2 regulation of microtubule dynamics

**Katelyn J Hoff[1], Jayne E Aiken[1,2], Mark A Gutierrez[2], Santos J Franco[2], Jeffrey K Moore[1]***

[1]Department of Cell and Developmental Biology, University of Colorado Anschutz Medical Campus, Aurora, United States; [2]Department of Pediatrics, University of Colorado Anschutz Medical Campus, Aurora, United States

***For correspondence:**
Jeffrey.moore@cuanschutz.edu

**Competing interest:** The authors declare that no competing interests exist.

**Abstract** Heterozygous, missense mutations in α- or β-tubulin genes are associated with a wide range of human brain malformations, known as tubulinopathies. We seek to understand whether a mutation's impact at the molecular and cellular levels scale with the severity of brain malformation. Here, we focus on two mutations at the valine 409 residue of TUBA1A, V409I, and V409A, identified in patients with pachygyria or lissencephaly, respectively. We find that ectopic expression of *TUBA1A*-V409I/A mutants disrupt neuronal migration in mice and promote excessive neurite branching and a decrease in the number of neurite retraction events in primary rat neuronal cultures. These neuronal phenotypes are accompanied by increased microtubule acetylation and polymerization rates. To determine the molecular mechanisms, we modeled the V409I/A mutants in budding yeast and found that they promote intrinsically faster microtubule polymerization rates in cells and in reconstitution experiments with purified tubulin. In addition, V409I/A mutants decrease the recruitment of XMAP215/Stu2 to plus ends in budding yeast and ablate tubulin binding to TOG (tumor overexpressed gene) domains. In each assay tested, the *TUBA1A*-V409I mutant exhibits an intermediate phenotype between wild type and the more severe *TUBA1A*-V409A, reflecting the severity observed in brain malformations. Together, our data support a model in which the V409I/A mutations disrupt microtubule regulation typically conferred by XMAP215 proteins during neuronal morphogenesis and migration, and this impact on tubulin activity at the molecular level scales with the impact at the cellular and tissue levels.

## Editor's evaluation

Tubulin mutations underlie a number of neurodevelopmental diseases, but their molecular effects remain largely unknown. Using a combination of approaches and model systems, Hoff et al., provide evidence that disease-associated α-tubulin mutations increase microtubule polymerization rates and block the recruitment of a regulatory binding protein, which combinatorially promotes excessive neurite branching that disrupts neuronal migration. Overall, this study demonstrates a link between the regulation of microtubule dynamics and disease pathogenesis.

## Introduction

Brain development requires cargo transport, force generation, and structural reinforcement by the microtubule cytoskeleton. The regulation of microtubules must be finely tuned to meet specific

**eLife digest** Proteins are molecules made up of long chains of building blocks called amino acids. When a mutation changes one of these amino acids, it can lead to the protein malfunctioning, which can have many effects at the cell and tissue level. Given that human proteins are made up of 20 different amino acids, each building block in a protein could mutate to any of the other 19 amino acids, and each mutations could have different effects.

Tubulins are proteins that form microtubules, thin tubes that help give cells their shape and allow them to migrate. These proteins are added or removed to microtubules depending on the cell's needs, meaning that microtubules can grow or shrink depending on the situation. Mutations in the tubulin proteins have been linked to malformations of varying severities involving the formation of ridges and folds on the surface of the brain, including lissencephaly, pachygyria or polymicrogyria.

Hoff et al. wanted to establish links between tubulin mutations and the effects observed at both cell and tissue level in the brain. They focused on two mutations in the tubulin protein TUBA1A that affect the amino acid in position 409 in the protein, which is normally a valine. One of the mutations turns this valine into an amino acid called isoleucine. This mutation is associated with pachygyria, which leads to the brain developing few ridges that are broad and flat. The second mutation turns the valine into an alanine, and is linked to lissencephaly, a more severe condition in which the brain develops no ridges, appearing smooth.

Hoff et al. found that both mutations interfere with the development of the brain by stopping neurons from migrating properly, which prevents them from forming the folds in the brain correctly. At the cellular level, the mutations lead to tubulins becoming harder to remove from microtubules, making microtubules more stable than usual. This results in longer microtubules that are harder for the cell to shorten or destroy as needed. Additionally, Hoff et al. showed that the mutant versions of TUBA1A have weaker interactions with a protein called XMAP215, which controls the addition of tubulin to microtubules. This causes the microtubules to grow uncontrollably.

Hoff et al. also established that the magnitude of the effects of each mutation on microtubule growth scale with the severity of the disorder they cause. Specifically, cells in which TUBA1A is not mutated have microtubules that grow at a normal rate, and lead to typical brain development. Meanwhile, cells carrying the mutation that turns a valine into an alanine, which is linked to the more severe condition lissencephaly, have microtubules that grow very fast. Finally, cells in which the valine is mutated to an isoleucine – the mutation associated with the less severe malformation pachygyria – have microtubules that grow at an intermediate rate.

These findings provide a link between mutations in tubulin proteins and larger effects on cell movement that lead to brain malformations. Additionally, they also link the severity of the malformation to the severity of the microtubule defect caused by each mutation. Further work could examine whether microtubule stabilization is also seen in other similar diseases, which, in the long term, could reveal ways to detect and treat these illnesses.

demands for different cell types, time points in development, and even different locations within a cell. Particularly in neurons, the spatial and temporal regulation of microtubules establishes cargo transport networks, facilitates efficient signal transduction, and supports the extension and retraction of neurites (*Dent and Baas, 2014*; *Dent and Kalil, 2001*; *He et al., 2002*; *Lin et al., 2012*; *Witte et al., 2008*). Accordingly, defects in microtubule regulation in neurons have been linked to brain malformations such as lissencephaly, microcephaly, and autism spectrum disorders, among others (*Bahi-Buisson et al., 2014*; *Chakraborti et al., 2016*; *Srivastava and Schwartz, 2014*).

Brain malformations such as lissencephaly, pachygyria, and polymicrogyria are associated with defects in neuronal migration. During neuronal migration, neurons that are born at the neuroepithelium must migrate out of the ventricular zone along radial glia cells to reach their destination at the cortical plate (*Barkovich et al., 2012*; *Barkovich et al., 2005*). Throughout the course of radial migration, cortical neurons must transition through various morphologies, first by transitioning from a bipolar to a multipolar state, then back to a bipolar state to complete migration (*Nadarajah et al., 2001*; *Noctor et al., 2004*; *Tabata and Nakajima, 2003*). After arriving at the cortical plate, the neurons are correctly positioned and polarized to connect to each other and send and receive information. During

this dynamic process of neuronal morphogenesis, microtubules generate force and provide structural support. Therefore, microtubule networks must be acutely regulated during these transitions (*Chesta et al., 2014*; *Dehmelt et al., 2006*; *Dent et al., 2011*; *He et al., 2002*; *Lu et al., 2013*; *Sainath and Gallo, 2015*; *Winding et al., 2016*). For example, the microtubule motor, kinesin-6, is important for establishing a leading process that is required for the transition from a multipolar to bipolar state and subsequent neuronal migration (*Falnikar et al., 2013*). Together this suggests the critical importance of regulating microtubules in the right place and at the right time during the morphological transitions neurons must undergo for migration and, ultimately, proper brain development.

'Tubulinopathies' encompass a wide range of heterozygous, missense mutations in the genes encoding α- and β-tubulins that are associated with a spectrum of brain malformations (*Bahi-Buisson et al., 2014*; *Fallet-Bianco et al., 2014*). α- and β-tubulins are encoded by a number of different genes, known as isotypes, that differ in amino acid sequence. The expression of different isotypes varies across different cell types as well as during different stages of development. Tubulinopathy mutations have been identified in three of the eight β-tubulin isotypes, and in one of the seven α-tubulin isotypes, a gene known as *TUBA1A* (*Bahi-Buisson et al., 2014*; *Ludueña and Banerjee, 2008*). *TUBA1A* is the most highly expressed α-tubulin isotype in post-mitotic neurons in the developing brain (*Buscaglia et al., 2020b*; *Gloster et al., 1994*; *Gloster et al., 1999*). To date, a total of 121 heterozygous, missense mutations have been identified in *TUBA1A* and are associated with neurodevelopment disorders (*Hebebrand et al., 2019*). It remains unclear whether the different severities of malformations observed in these patients are a result of differences in genetic background or are a result of a specific functional difference in the mutant tubulin. A few of these *TUBA1A* mutants have been identified as loss-of-function mutants that are unable to properly assemble into microtubule polymer and thus result in an undersupply of tubulin in the cell (*Belvindrah et al., 2017*; *Keays et al., 2007*). However, other *TUBA1A* mutants appear to be gain-of-function, as they are capable of microtubule assembly and act dominantly to perturb microtubule function in migrating neurons (*Aiken et al., 2020*; *Aiken et al., 2019*). If haploinsufficiency does not explain all *TUBA1A* tubulinopathies, then it is necessary to define clear mechanistic models for how different mutations in α-tubulin can result in different developmental outcomes. Only a small fraction of *TUBA1A* mutants have been studied, and more continue to be identified in the clinic. Therefore, it is important to establish clear mechanistic themes that will be useful in predicting the molecular, cellular, and tissue-level phenotypes of specific mutations.

The fundamental role of α-tubulin is to complex with β-tubulin into heterodimers and form microtubule polymers that are dynamic and highly regulated via both intrinsic and extrinsic factors (*Bodakuntla et al., 2019*; *Borys et al., 2020*; *Goodson and Jonasson, 2018*; *Manka and Moores, 2018*; *Mitchison and Kirschner, 1984*). Purified tubulin assembles into dynamic microtubules in the absence of extrinsic microtubule-associated proteins (MAPs) that are found in cells (*Horio and Hotani, 1986*; *Mitchison and Kirschner, 1984*; *Walker et al., 1988*). This intrinsic activity is in part regulated by the series of conformational states that tubulin undergoes as it transitions from free heterodimer into microtubule polymer (*Chrétien et al., 1995*; *Mandelkow et al., 1991*). In recent years, studies using cryo-electron microscopy and recombinant tubulin approaches have advanced the field's understanding of the transitions that accompany GTP hydrolysis as free tubulin assembles into microtubule lattice (*Geyer et al., 2018*; *Manka and Moores, 2018*; *Roostalu et al., 2020*; *Zheng et al., 1995*). In addition to nucleotide-dependent conformational changes, tubulin must transition from a 'curved' free heterodimer to a straight conformation in the microtubule lattice (*Buey et al., 2006*; *Jánosi et al., 1998*; *Nawrotek et al., 2011*; *Nogales and Wang, 2006*; *Ravelli et al., 2004*; *Rice et al., 2008*). Tubulin domains that control the curved-to-straight transition are not completely understood. The ability of tubulin to adopt a range of conformational states not only impacts its intrinsic microtubule dynamics, but also tubulin's interactions with a wide spectrum of MAPs. For example, doublecortin (DC) is predominantly expressed in the developing brain and its N-terminal DC-like repeat domain binds at the corner of four tubulin subunits in a straight microtubule lattice (*Fourniol et al., 2010*; *Manka and Moores, 2020*; *Moores et al., 2006*). On the other hand, members of the XMAP215 protein family possess a variable number of TOG (tumor overexpressed gene) domains, some of which preferentially bind to curved tubulin (*Ayaz et al., 2014*; *Ayaz et al., 2012*; *Slep and Vale, 2007*). XMAP215 proteins use this activity to either pull curved tubulin toward the microtubule plus end for polymerization or drive lattice-bound tubulin into a curved state and promote catastrophe

(*Ayaz et al., 2014*; *Brouhard et al., 2008*; *Farmer et al., 2021*; *Geyer et al., 2018*). Therefore, understanding the effects of tubulin conformations on both intrinsic activity and extrinsic regulatory mechanisms are essential for elucidating the regulation of microtubule dynamics.

In this study we investigate the molecular mechanism of two previously identified tubulinopathy-associated mutations at the valine 409 residue of *TUBA1A*, V409I, and V409A. The patient harboring the V409A mutation exhibited lethal lissencephaly, whereas the patient with the V409I mutation presents with a milder malformation known as pachygyria (*Bahi-Buisson et al., 2014*; *Fallet-Bianco et al., 2014*). Studying these two mutations that occur at the same residue, yet are associated with varying degrees of brain malformations in patients, has potential to provide insight into how perturbing tubulin activity and the microtubule cytoskeleton in slightly different ways ultimately impacts brain development. We find that the expression of V409I or V409A dominantly disrupts neuron migration and morphology in mice and neuron branching during in vitro development. Additionally, V409I/A mutants increase microtubule acetylation and polymerization rates in neuronal cultures. By modeling the V409I/A mutants at the corresponding residue in budding yeast, V410, we find that the mutants increase microtubule polymerization and depolymerization rates in cells and disrupt XMAP215/Stu2 activity. Furthermore, the V410A mutation promotes faster microtubule polymerization in reconstitution experiments with purified tubulin, indicating a change in tubulin's intrinsic polymerization activity. We propose that these results may be reconciled by the V410I/A mutants adopting a persistently straightened state as compared to wild-type (WT) tubulin, and this drives intrinsically faster microtubule polymerization rates in vitro. Importantly, we find that the severity of the V409I and V409A mutations scale from protein activity, to microtubule dynamics in cells, to neuron morphology and migration. Together, our results suggest that the *TUBA1A*-V409I/A mutations cause brain malformations by subverting dynamic regulation typically conferred by XMAP215.

## Results

### *TUBA1A*-V409I/A mutants dominantly disrupt cortical neuron migration and morphology

The V409 residue of *TUBA1A* is located in helix 11′ of α-tubulin, near the intradimer interface on the cytoplasmic surface of the microtubule (*Alushin et al., 2014*; PDB: 3J6F; *Figure 1—figure supplement 1A*). To determine whether *TUBA1A*-V409I/A variants act as loss- or gain-of-function mutations, we first sought to establish whether the mutant proteins can polymerize into microtubules. We expressed a hexahistidine (6X-His)-tagged *TUBA1A* construct in cortical neuron cultures harvested from P0 rats. The 6X-His tag was inserted between residues I42 and G43 in a flexible loop of *TUBA1A* that allows for the addition of amino acids without perturbing tubulin function (*Buscaglia et al., 2020a*). After 2 days in culture (days in vitro 2 [DIV2]), we extracted soluble tubulin from the neurons, fixed the cells, and stained for the 6X-His tag. Both the *TUBA1A*-V409I and -V409A mutants incorporate into microtubules, similar to WT controls, as indicated by the microtubule filaments that can be resolved in the soma (*Figure 1A*). As a positive control, we expressed the *TUBA1A*-T349E mutant that does not assemble into polymer, as evidenced by the lack of labeled polymer in the soma (*Figure 1A*). We find that the amount of 6X-His signal retained after soluble protein extraction is similar between WT- and V409I/A-expressing cells (*Figure 1B*). These results suggest that the *TUBA1A*-V409I/A mutants assemble similar levels of ectopically expressed tubulin into microtubule polymer as compared to WT.

We next asked whether incorporation of the *TUBA1A*-V409I/A mutants into microtubules in vivo is sufficient to disrupt cortical neuron migration. To answer this question, we performed in utero electroporations to ectopically express *TUBA1A* in migrating neurons in the developing mouse brain. At embryonic day (E) 14.5, we electroporated neural progenitors in the cortical ventricular zone, which give rise to immature excitatory neurons that migrate from the ventricular zone into the upper cortical plate. The cDNA expression plasmids used for the electroporations encode for either WT or mutant *TUBA1A*, along with GFP to identify the electroporated cells. The majority of GFP + neurons ectopically expressing *TUBA1A*-WT successfully migrate from the ventricular zone to the upper cortical plate by E18.5 (*Figure 1C*). In contrast, neurons ectopically expressing *TUBA1A*-V409I do not migrate to the upper cortical plate as robustly as WT control cells by E18.5, as evidenced by many neurons remaining in the ventricular, subventricular, and intermediate zones. Strikingly, a majority of *TUBA1A*-V409A-expressing neurons fail to migrate at all by E18.5 and remain primarily in the ventricular and

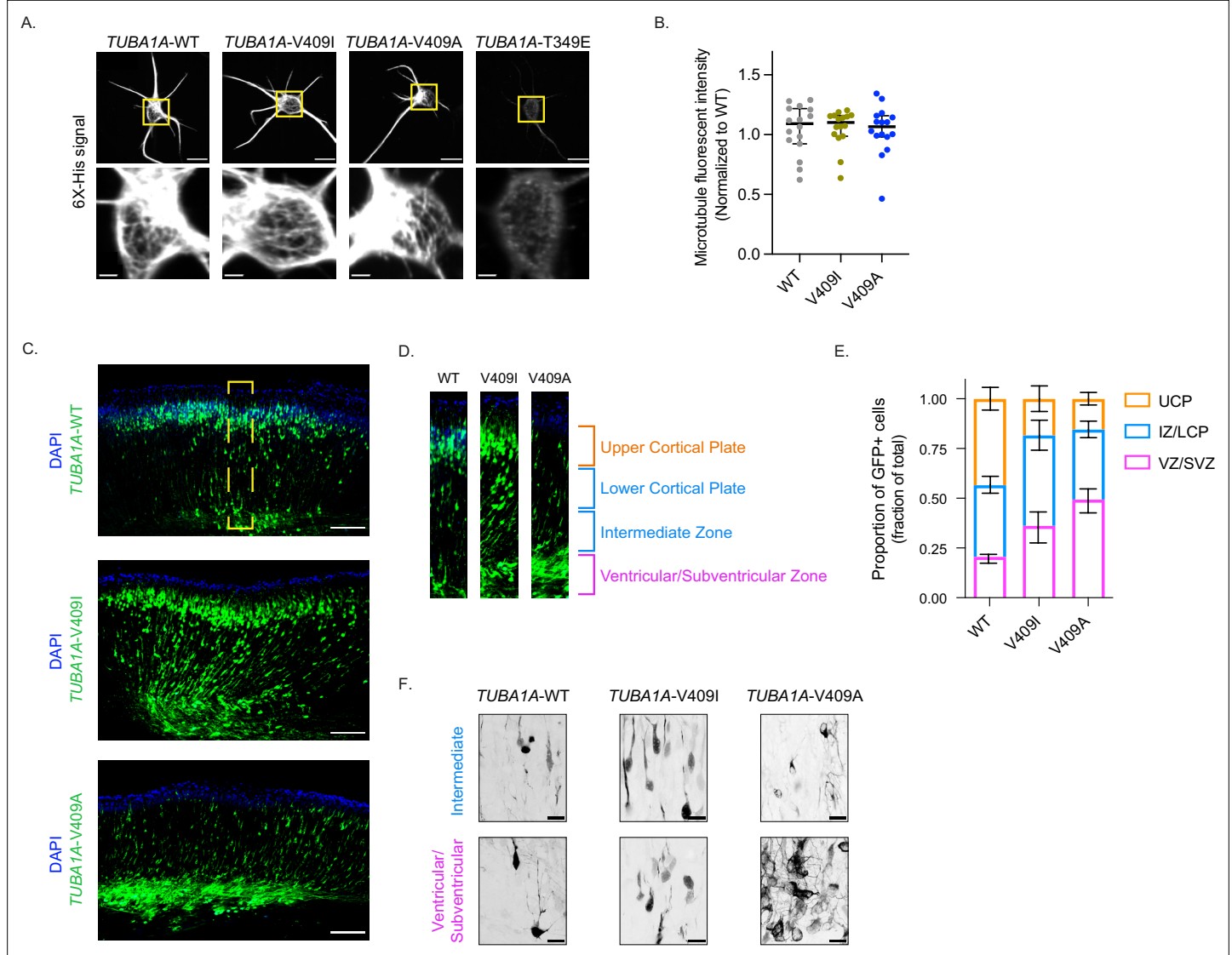

**Figure 1.** *TUBA1A*-V409I/A mutants dominantly disrupt cortical neuron migration. (**A**) Representative images of neurons ectopically expressing 6X-His-tagged *TUBA1A*-WT, -V409I, -V409A, and -T349E. The soluble tubulin was extracted from the cells, and then cells were stained with an anti-6X-His antibody. Bottom insets are representative of yellow boxes. Scale bar = 10 μm in top panels. Scale bar = 2 μm in bottom panels. Images were collected from three independent experiments; n=16 cells for wild-type (WT), n=16 for V409I, and n=16 for V409A. (**B**) 6X-His fluorescence intensity was measured for each neuron imaged in panel A. Each dot represents a single cell. Bars are median ± 95% confidence interval. (**C**) Representative coronal sections of E18.5 mouse brains that were electroporated with the above constructs at E14.5. GFP labels electroporated cells. Scale bar = 200 μm. Images were collected from three independent experiments; n=8 sections for WT, n=10 for V409I, and n=14 for V409A. Yellow box represents region in (**D**). (**E**) Representative cortical region was divided into four equal quartiles and GFP fluorescence intensity was measured in each. 1st quartile = ventricular/subventricular zones (VZ/SVZ), 2nd quartile = intermediate zone (IZ), 3rd quartile = lower cortical plate (LCP), 4th quartile = upper cortical plate (UCP). Bars are mean ± 95% confidence interval. (**F**) Representative zoomed in images of cells in the VZ/SVZ and the IZ. Scale bar = 20 μm. Images were collected from same set of coronal sections described in (**C**).

The online version of this article includes the following source data and figure supplement(s) for figure 1:

**Source data 1.** 6X-His tagged tubulin fluorescence intensity.

**Source data 2.** Proportion of GFP+ cells in quartiles of cortical sections.

**Figure supplement 1.** Effects of human *TUBA1A*-V409I/A mutants.

subventricular zones. To quantify migration for comparison across experiments, we divided the cortical sections into four equal segments from the ventricular zone to the top of the cortical plate and measured the proportion of GFP signal in each segment (*Figure 1D*). Approximately, the first quartile represents the ventricular and subventricular zones, the second quartile represents the intermediate zone, the third quartile represents the lower cortical plate, and the fourth quartile represents the upper cortical plate. We find that the largest proportion of *TUBA1A*-WT-expressing cells (43.4%) are found in the upper cortical plate region (labeled in orange), while the largest proportion of *TUBA1A*-V409I-expressing cells (45.4%) are in the intermediate zone and lower cortical plate, and the largest proportion of *TUBA1A*-V409A-expressing cells (49.4%) are in the ventricular/subventricular zones (*Figure 1E* and *Figure 1—figure supplement 1B*). In each of our experiments, untransfected control cells successfully reach the cortical plate, as evidenced by abundant DAPI signal in the upper cortical plate segment, indicating that the radial glia cells that support migration are not impaired by the transfection of mutant *TUBA1A* in our experiments. These results suggest that the *TUBA1A*-V409I/A mutants act dominantly to disrupt neuron migration, with *TUBA1A*-V409A being more severe than the *TUBA1A*-V409I mutant.

To further our understanding of how this migration defect occurs, we next used high-resolution confocal microscopy to examine the morphologies of cells that remain in the intermediate and ventricular zones 4 days after transfection. We find that the *TUBA1A*-WT-expressing cells in the intermediate and ventricular zones exhibit a clear bipolar morphology (*Figure 1F*). *TUBA1A*-V409I-expressing cells in the intermediate and ventricular zones also show a primarily bipolar morphology, however it is again evident that there are more *TUBA1A*-V409I-expressing cells in these regions as compared to WT-expressing cells (*Figure 1F*). Most strikingly, the *TUBA1A*-V409A-expressing cells in the intermediate and ventricular zones have elaborate multipolar morphologies with many processes and branches (*Figure 1F*). These complex morphologies and the high density of *TUBA1A*-V409A-expressing cells in these regions create a tangled nest of processes that make it difficult to measure the exact number of processes per cell. We conclude that *TUBA1A*-V409A-expressing cells exhibit severe migration defects accompanied by morphological defects, while *TUBA1A*-V409I-expressing cells exhibit intermediate migration defects without a strong morphological phenotype.

## *TUBA1A*-V409I/A mutants increase neurite branching and decrease neurite retraction events

The microtubule cytoskeleton plays a crucial role in the series of morphological changes that cortical neurons must undergo throughout the course of radial migration (*Nadarajah et al., 2001*; *Noctor et al., 2004*; *Tabata and Nakajima, 2003*). When immature neurons are born, they extend numerous neurites to probe their environment for directional cues. Once these cues have been identified, neurons must retract most of their neurites and become bipolar, such that one neurite becomes the axon and a neurite on the opposite side of the cell becomes the leading process that guides radial migration. Therefore, it is particularly crucial that microtubules are able to deftly respond to various cues throughout development that promote these different morphology transitions. Based on the highly branched morphologies of V409A neurons in the ventricular and intermediate zones (*Figure 1F*), we hypothesized that V409 mutants could either (1) initiate ectopic neurite growth, (2) promote faster neurite growth, or (3) inhibit or slow neurite retraction.

To distinguish between these hypotheses, we examined the in vitro development of cultured primary neurons ectopically expressing the 6X-His-tagged *TUBA1A* (WT or mutant) plasmids described above. We first compared the number of primary, secondary, and tertiary neurites at two morphologically distinct stages of in vitro cortical neuron development, stage 2 (DIV1) and stage 3 (DIV2). Stage 2 neurons are characterized as being multipolar with all neurites being approximately similar in length (*Dotti et al., 1988*). On average, there is no appreciable difference in the number of neurites quantified in stage 2 cells expressing WT or V409I/A mutant *TUBA1A*. Most neurons at stage 2 have four primary neurites, and we rarely observe secondary or tertiary branches (*Figure 2—figure supplement 1A* and B). Consistent with this finding, stage 3 WT and V409I/A mutant neurons have a similar number of primary neurites (5.1±0.4, 4.9±0.4, 5.4±0.4 for WT, V409I, and V409A, respectively; *Figure 2A and B*). However, *TUBA1A*-V409I-expressing cells exhibit a slight, but measurable increase in secondary branches as compared to WT cells (0.6±0.2, 0.9±0.3 branches/cell for WT and V409I, respectively; *Figure 2B*). Strikingly, *TUBA1A*-V409A-expressing cells have a significantly greater

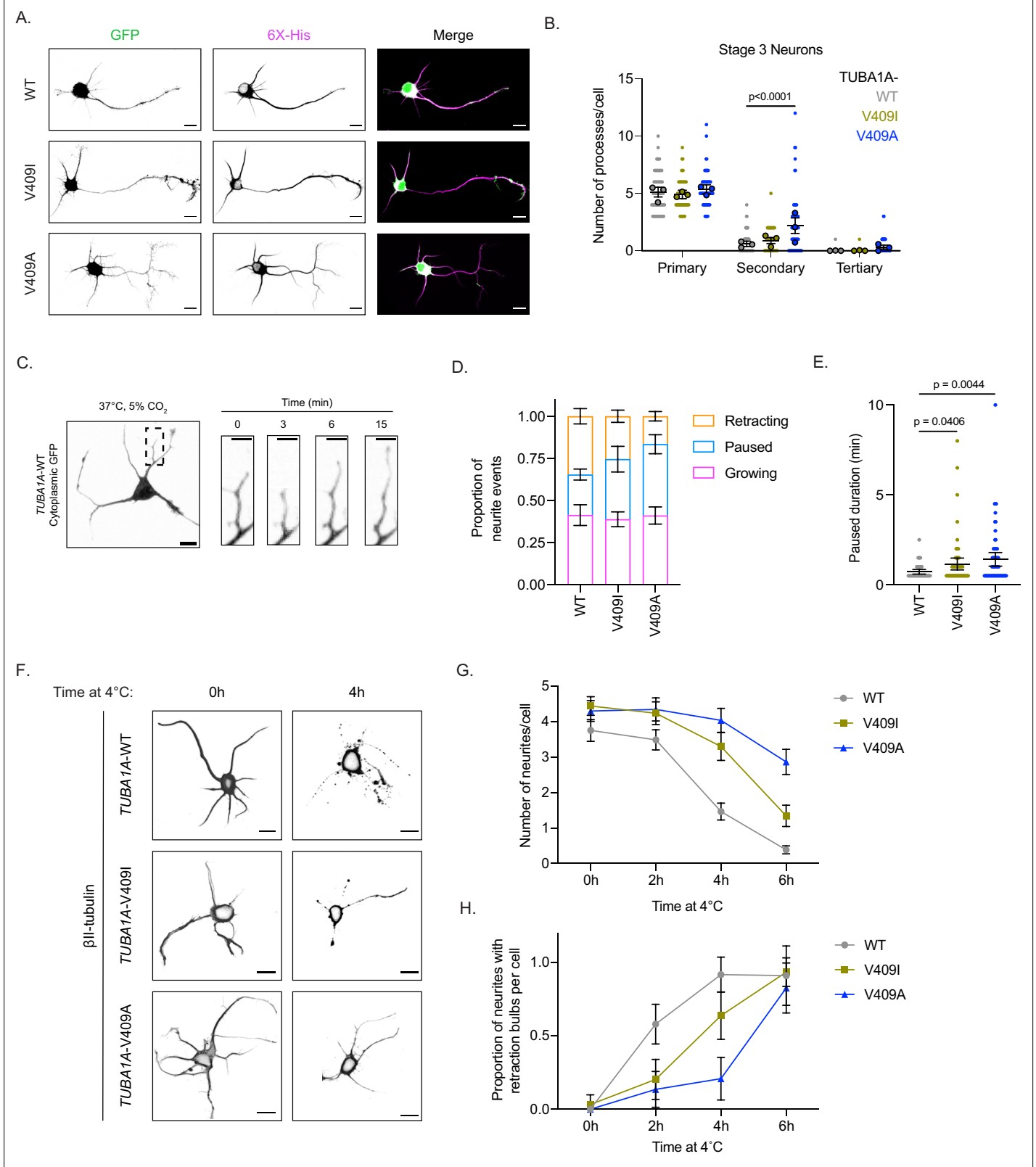

**Figure 2.** *TUBA1A*-V409I/A mutants alter neuron morphologies. (**A**) Representative days in vitro 2 (DIV2) cortical neurons expressing cytoplasmic GFP and 6X-His-tagged *TUBA1A*-WT, -V409I, and -V409A. Scale bar = 10 µm. (**B**) Quantification of number of primary, secondary, and tertiary branches along axons and dendrites in each condition. Images were collected from three independent experiments; n=65 cells for wild-type (WT), n=56 for V409I, and n=59 for V409A. Each dot represents a single cell. Bars are mean ± 95% confidence interval. Statistical analysis between multiple groups was analyzed

*Figure 2 continued on next page*

*Figure 2 continued*

by two-way ANOVA and corrected for multiple comparisons post hoc by Tukey test. All statistics with p≤0.05 are indicated on graph. (**C**) Example image of DIV2 cortical neuron expressing cytoplasmic GFP and *TUBA1A*-WT. Scale bar = 10 μm. Yellow box indicates a representative region measured every 30 s for 15 min at 37°C, 5% $CO_2$. Scale bar in insets = 5 μm. (**D**) Proportion of events classified as the neurite growing, paused, or shrinking, depending on if the length of the neurite grew, stayed the same, or shrunk, respectively. Images were collected from three independent experiments; n=10 cells for WT, n=13 for V409I, and n=13 for V409A. Bars are mean ± standard error of the mean. (**E**) Quantification of the amount of time a neurite was in a paused state, neither growing nor shrinking. Each dot represents a period of pause; cell numbers are the same as stated in (**D**). Bars are mean ± 95% confidence interval. Statistical analysis was done using an unpaired t-test. All statistics with p≤0.05 are indicated on graph. (**F**) Representative images of neurons expressing the above plasmids, exposed to 4°C for indicated time, and stained with TUBB2A/B. Scale bar = 10 μm. (**G**) Quantification of the number of neurites per cell every 2 hr over a 6 hr 4°C cold shock. Images were collected from three independent experiments; for WT n=36 cells at 0 hr, n=35 at 2 hr, n=41 at 4 hr, n=44 at 6 hr; for V409I n=32 at 0 hr, n=30 at 2 hr, n=27 at 4 hr, n=29 at 6 hr; for V409A n=37 at 0 hr, n=29 at 2 hr, n=27 at 4 hr, n=22 at 6 hr. Dots represent averages from the three separate experiments and error bars are ± 95% confidence interval. (**H**) The proportion of neurites that have retraction bulbs per cell measured every 2 hr over a 6 hr 4°C cold shock. Cell numbers and error bars are the same as stated in (**G**).

The online version of this article includes the following video, source data, and figure supplement(s) for figure 2:

**Source data 1.** Number of processes per cell.

**Source data 2.** Neurite dynamics under standard conditions.

**Source data 3.** Number of neurites and retraction bulbs under cold temperature conditions.

**Figure supplement 1.** Effect of *TUBA1A*-V409I/A on neuronal morphologies.

**Figure 2—video 1.** Neurite dynamics.

https://elifesciences.org/articles/76189/figures#fig2video1

number of secondary branches, and some also exhibit tertiary branches (secondary: 0.6±0.2, 2.2±0.7 branches/cell, tertiary: 0.01±0.03, 0.3±0.2 branches/cell for WT and V409A, respectively; *Figure 2B*). These data fail to support the hypothesis that *TUBA1A*-V409I/A mutants initiate ectopic primary neurite growth because we find a similar number of primary neurites in WT and mutant-expressing cells. Our data instead lend support to the alternative hypotheses that there is either faster neurite growth or insufficient neurite retraction, as evidenced by the increase in neurite branching.

To assess potential differences in neurite extension or retraction, we imaged living neurons expressing *TUBA1A*-WT, -V409I, or -V409A over time to measure neurite dynamics (*Figure 2C*; *Figure 2—video 1*). We find that neurite growth rates and retraction rates are similar between *TUBA1A*-WT-, -V409I-, and -V409A-expressing cells (*Table 1*). The duration of growth and retraction events is also not significantly different; however, neurites in V409I-expressing cells, and more strikingly V409A, have a lower proportion of retraction events per neurite as compared to WT (34.6%, 25.4%, and 16.6% for WT, V409I, and V409A, respectively; *Figure 2D* and *Figure 2—figure supplement 1C*). In addition, neurites in both mutants commonly exhibit a paused state, where they are neither growing nor shrinking. Both the proportion of paused events and the duration of these pauses are significantly increased in V409I/A mutants (proportions: 24.0%, 35.6%, and 42.3%, duration: 0.7, 1.2, and 1.4 min for WT, V409I, and V409A, respectively; *Figure 2D and E*; *Figure 2—figure supplement 1C*; *Table 1*). These data suggest that the increased branching observed in V409I/A neurons is not due to an increase in neurite growth events or rate. Rather, our data supports the hypothesis that V409I/A cells have fewer retraction events and spend an increased amount of time neither growing nor shrinking.

We next asked whether the observed differences in neurite dynamics are attributable to increased microtubule stability in V409I/A mutants, compared to WT. We predicted that V409I/A-expressing cells may maintain neurites under microtubule destabilizing conditions that are sufficient to stimulate neurite loss in WT controls. We stimulated neurite loss in vitro by placing the neuron cultures at 4°C over the course of 6 hr. At low temperatures, microtubules destabilize, which ultimately leads to neurite retraction (*Breton and Brown, 1998*; *Tilney and Porter, 1967*; *Weber et al., 1975*). During

**Table 1.** Neurite dynamics.

|  | Growth rate (μm/min) | Retraction rate (μm/min) | Growth duration (min) | Retraction duration (min) | Paused duration (min) |
|---|---|---|---|---|---|
| WT | 2.7±0.4 | 2.8±0.5 | 0.9±0.1 | 0.8±0.1 | 0.7±0.1 |
| V409I | 2.4±0.2 | 2.4±0.4 | 0.9±0.1 | 0.6±0.1 | 1.2±0.3 |
| V409A | 2.7±0.3 | 2.4±0.6 | 0.8±0.1 | 0.7±0.1 | 1.4±0.4 |

our time course experiment, cells expressing *TUBA1A*-WT, -V409I, or -V409A were fixed at 2-hr intervals and stained for β-II tubulin, a highly abundant β-tubulin isotype in cortical neurons, to label microtubules in neurites. After 4 hr at 4° C, the average number of neurites in *TUBA1A*-WT-expressing cells (identified by cytoplasmic GFP displayed in *Figure 2—figure supplement 1C*) is decreased from 4 to 1.5. The remaining neurites are depleted for β-II tubulin signal and nearly all have the retraction bulbs that are characteristic of retracting neurites (*Figure 2F–H* and *Figure 2—figure supplement 1D-F*). By 6 hr at 4°C, the number of neurites per cell is further diminished and β-II tubulin signal is only found in the cell soma of neurons expressing *TUBA1A*-WT (*Figure 2—figure supplement 1D*). In contrast to WT controls, neurite loss and the accumulation of retraction bulbs is more gradual in *TUBA1A*-V409I-expressing cells (*Figure 2F–G*). The most severe effect is detected in *TUBA1A*-V409A-expressing cells where neurites containing robust β-II tubulin signal are maintained throughout the time course, and retraction bulbs begin to form only after 6 hr at 4°C (*Figure 2G–H*). Together, these data suggest that expression of *TUBA1A*-V409I or -V409A creates hyper-stable microtubules in neurites that decrease neurite retraction events and lead to excessive branching.

## *TUBA1A*-V409I/A mutants alter microtubule dynamics in neurons

We next sought to determine if the increase in neurite branching and resistance to cold-induced microtubule destabilization seen in *TUBA1A*-V409I/A-expressing cells is a result of altered microtubule dynamics. To test this, we first quantified levels of tubulin post-translational modifications (PTMs) in WT- and mutant-expressing cells. Tyrosinated tubulin is a marker of dynamic, newly formed microtubules and is typically evenly distributed among the axon and dendrites (reviewed in *Westermann and Weber, 2003*). Acetylated tubulin is a marker of stable microtubule polymer and is highly localized to the axon, but largely absent in the dendrites.

As expected, stage 3 control cells ectopically expressing *TUBA1A*-WT have acetylated tubulin localized primarily to the axon and tyrosinated tubulin throughout the neuron (*Figure 3A and B*). Additionally, *TUBA1A*-WT cells have the characteristic peak of tyrosinated tubulin at the distal tip of the axon, followed by a sharp increase in acetylated tubulin in the more proximal regions of the process (*Figure 3—figure supplement 1A*). Both *TUBA1A*-V409I- and -V409A-expressing neurons also exhibit tyrosinated tubulin throughout the neuron, similar to that observed in control cells expressing *TUBA1A*-WT (*Figure 3B*, *Figure 3—figure supplement 1B-D*). *TUBA1A*-V409I-expressing cells have a modest, yet statistically insignificant increase in microtubule acetylation at the distal axon tip, and a significant increase in microtubule acetylation in the region more proximal to the soma (*Figure 3C*, *Figure 3—figure supplement 1B*). *TUBA1A*-V409A cells exhibit a striking increase in microtubule acetylation at the distal axon tip, as well as in the more proximal region of the process (first 10 µm from distal tip: 1.238±0.303, 1.801±0.372, 2.331±0.596 AU; second 10 µm: 2.442±0.416, 3.505±0.556, 3.868±0.702 AU for WT, V409I, an V409A, respectively; *Figure 3C*, *Figure 3—figure supplement 1C*). Comparing the levels of tyrosinated and acetylated tubulin in each cell, we find an increase in the ratio of acetylated to tyrosinated tubulin in the distal 20 µm of the axon in V409I/A cells as compared to WT (*Figure 3D*). In a similar analysis of the most distal 10 µm of dendrites, we find that there are no significant differences in acetylated tubulin levels between WT and V409I/A cells, and thus no difference in the ratio of acetylated to tyrosinated tubulin (*Figure 3—figure supplement 1E-F*). However, we observed that some V409I/A cells exhibit increased microtubule acetylation in the dendrite regions that are closer to the soma, which was not observed in WT-expressing cells (*Figure 3A*). These data suggest that cells expressing V409I/A mutants have increased levels of microtubule acetylation as compared to WT, and V409A cells in particular have abnormally high levels of acetylation at the distal axon tip.

We next asked whether the changes observed in microtubule acetylation are attributable to altered microtubule dynamics in these neurons. To test this we measured microtubule polymerization events in living cells by transfecting cultured cortical neurons with pCIG2 constructs that co-express *TUBA1A*-WT, -V409I, or -V409A along with GFP-MACF43 (*Figure 3E*; *Figure 3—video 1*; *Honnappa et al., 2009*). We find that the number of growing microtubule events that occur over a 4-min imaging experiment is similar across all genotypes (1846±229, 1746±217, 1785±178 polymerization events/cell/4 min for WT, V409I, V409A, respectively; *Figure 3F*). However, compared to WT, V409A-expressing cells have faster microtubule polymerization rates, while V409I cells have an intermediate phenotype between the two (5.2±0.1, 6.2±0.1, 6.4±0.1 µm/min for WT, V409I, V409A, respectively;

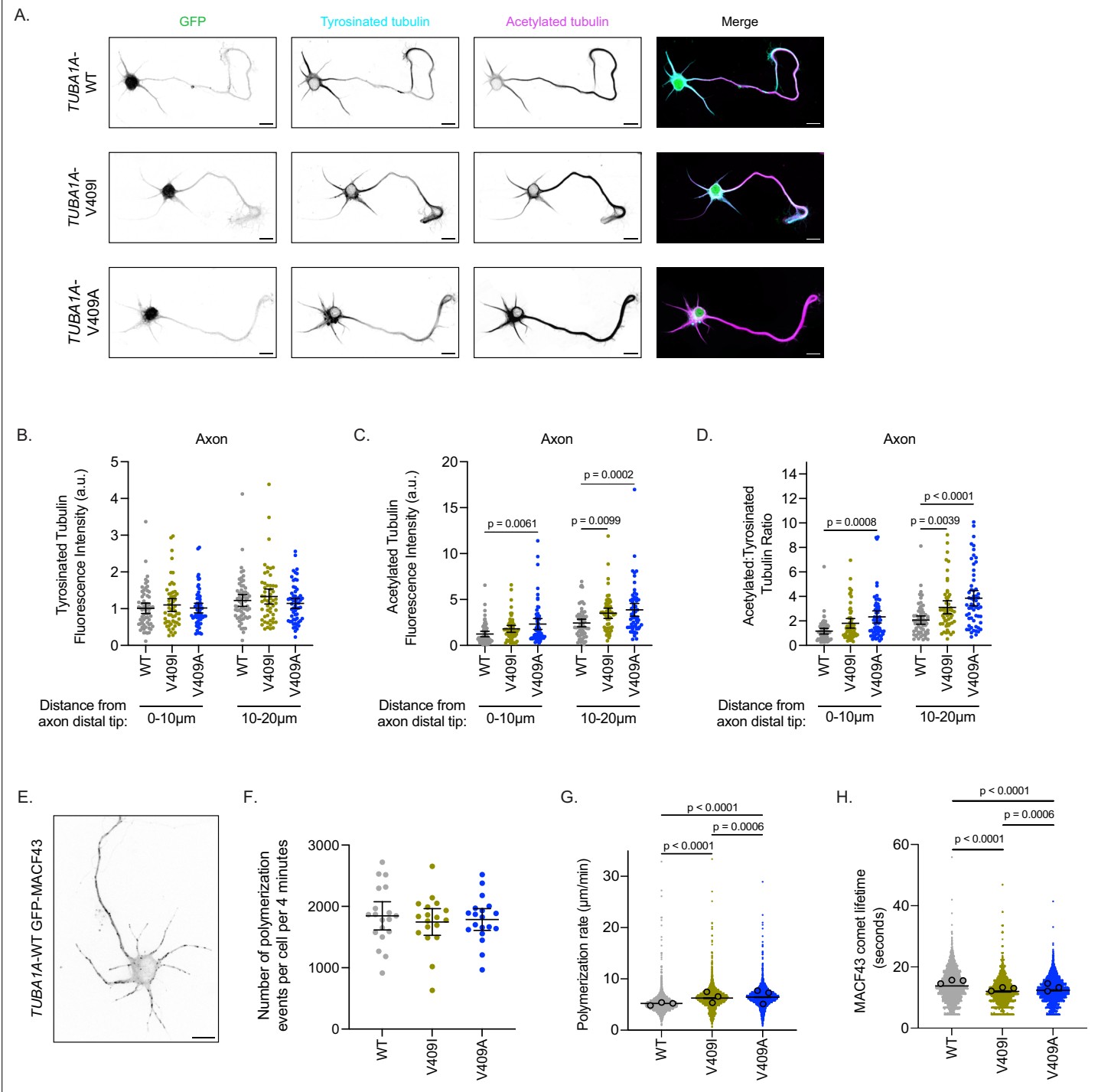

**Figure 3.** *TUBA1A*-V409I/A microtubules have altered microtubule dynamics in neurons. (**A**) Representative days in vitro 2 (DIV2) cortical neurons expressing plasmids with cytoplasmic GFP and 6X-His-tagged *TUBA1A*-WT, -V409I, and -V409A. Cells stained with tyrosinated tubulin and acetylated tubulin. Scale bar = 10 µm. (**B**) Quantification of acetylated tubulin fluorescence intensity or tyrosinated tubulin (**C**) binned in two, 10 µm segments at the distal end of the axon tip. Images were acquired in three independent experiments; n=61 cells for wild-type (WT), n=54 for V409I, n=59 for V409A. Each dot represents the quantification from one cell. Bars are mean ± 95% confidence interval. Statistical analysis between multiple groups was analyzed by two-way ANOVA and corrected for multiple comparisons post hoc by Tukey test. All statistics with p≤0.05 are indicated on graph. (**D**) Calculated ratio of acetylated tubulin over tyrosinated tubulin measurements displayed in (**B**) and (**C**). (**E**) Example image of DIV2 cortical neuron co-expressing *TUBA1A*-WT and GFP-MACF43. Scale bar = 10 µm. (**F**) Quantification of the number of microtubule polymerization events that occur in each cell over the course of 4 min. Images were obtained from three independent experiments; n=19 cells for WT, n=18 for V409I, n=19 for V409A. Each dot represents one cell. Bars represent mean ± 95% confidence interval. Statistical analysis for panels F–H was done using an unpaired t-test. All statistics with p≤0.05 are

*Figure 3 continued on next page*

*Figure 3 continued*

indicated on graph. (**G**) Quantification of microtubule polymerization rates measured from GFP-MACF43 comets. Each dot represents the rate of one microtubule polymerization rate. Larger dots outlined in black indicate the average of all polymerization rates from the three independent experiments where these images were obtained. (**H**) Quantification of GFP-MACF43 comet lifetime in seconds.

The online version of this article includes the following video, source data, and figure supplement(s) for figure 3:

**Source data 1.** Microtubule post-translational modifications fluorescence intensity.

**Source data 2.** Quantification of microtubule dynamics in neurons by GFP-MACF43 comets.

**Figure supplement 1.** *TUBA1A*-V409I/A microtubules have increased microtubule acetylation but not tyrosination.

**Figure 3—video 1.** Microtubule dynamics in neurons.

https://elifesciences.org/articles/76189/figures#fig3video1

*Figure 3G*). In addition, V409I and V409A cells have slightly but significantly shorter MACF43 comet lifetimes as compared to WT (13.8±0.2, 12.0±0.2, 12.4±0.2 s for WT, V409I, V409A, respectively; *Figure 3H*). We conclude that ectopic expression of *TUBA1A*-V409I/A mutants alters microtubule dynamics and microtubule acetylation in neurons.

## Modeling *TUBA1A*-V409I/A mutants in budding yeast reveals altered microtubule dynamics

The α-tubulin V409 residue is highly conserved across eukaryotes (*Figure 4A*). To better understand the molecular impact of the *TUBA1A*-V409I/A mutants, we created analogous mutants at the corresponding residue in *Saccharomyces cerevisiae* (or budding yeast), V410. Using this system, we created strains in which all the α-tubulin expressed in the cell was either WT, V410I, or V410A. We find that compared to WT, V410I cells have no significant change in fitness, while V410A cells exhibit a slight fitness defect, as indicated by a 4.4% increase in doubling time (*Figure 4—figure supplement 1A*). By introducing either the V410I or V410A mutation into both α-tubulin isotypes of budding yeast (*TUB1* and *TUB3*), we can measure the dynamics of individual microtubules that consist of a homogenous supply of either WT or mutant α-tubulin. Thus, any effect we see on dynamics would be a result of the mutant of interest as opposed to a compensatory response by alternative tubulin isotypes.

To track the length of individual microtubules over time in cells, we used Bik1-3GFP, the yeast homologue of CLIP-170, as a marker for microtubule plus ends (*Figure 4B and C*). Compared to WT, we find that *tub1/tub3*-V410A microtubules have significantly faster polymerization rates, while *tub1/tub3*-V410I microtubules have intermediate rates between WT and V410A mutants (1.1±0.1, 1.5±0.1, 1.6±0.1 μm/min for WT, V410I, and V410A, respectively; *Figure 4D*). Similarly, *tub1/tub3*-V410A microtubules, and to a lesser extent -V410I microtubules, have faster depolymerization rates compared to WT (1.9±0.2, 2.3±0.2, 2.8±0.3 μm/min for WT, V410I, V410A, respectively; *Figure 4E*). Our data also shows that *tub1/tub3*-V410A microtubules exhibit very few catastrophes compared to WT, and the *tub1/tub3*-V410I microtubules again have an intermediate catastrophe frequency (1.216±0.105, 0.973±0.111, 0.761±0.106 events/min for WT, V410I, and V410A, respectively; *Figure 4F*). Accordingly, *tub1/tub3*-V410A mutant microtubules reach longer median lengths than either the *tub1/tub3*-V410I or WT microtubules (0.66, 0.68, 0.90 μm for WT, V410I, V410A, respectively; *Figure 4G*). Additionally, we find that astral microtubules in *tub1/tub3*-V410A mutant cells are retained for a longer time at 4°C than those in WT control cells, and *tub1/tub3*-V410I microtubules have an intermediate phenotype (*Figure 4H*). Summarizing these dynamics data, we find that α-tubulin-V410I/A microtubules, and particularly the *tub1/tub3*-V410A variant, exhibit faster microtubule polymerization rates and decrease how often the microtubule catastrophes. However, when these mutant microtubules do catastrophe, they depolymerize at a faster rate than WT.

To understand how the different microtubule parameters described above work together to influence microtubule activity, we calculated the dynamicity of WT and *tub1/tub3*-V410I/A microtubules. Dynamicity is defined as the total change in microtubule length divided by the change in time (*Jordan et al., 1993*). We find that *tub1/tub3*-V410A microtubules have the highest dynamicity values, while *tub1/tub3*-V410I microtubules have an intermediate dynamicity between -V410A and WT (0.443±0.028, 0.507±0.034, 0.563±0.047 subunits/s; *Figure 4I*). Similar to our work in neurons, the V410A mutant has the strongest effect on microtubule dynamics while the V410I mutant has a more intermediate impact. Together, the results of modeling these patient-associated mutations

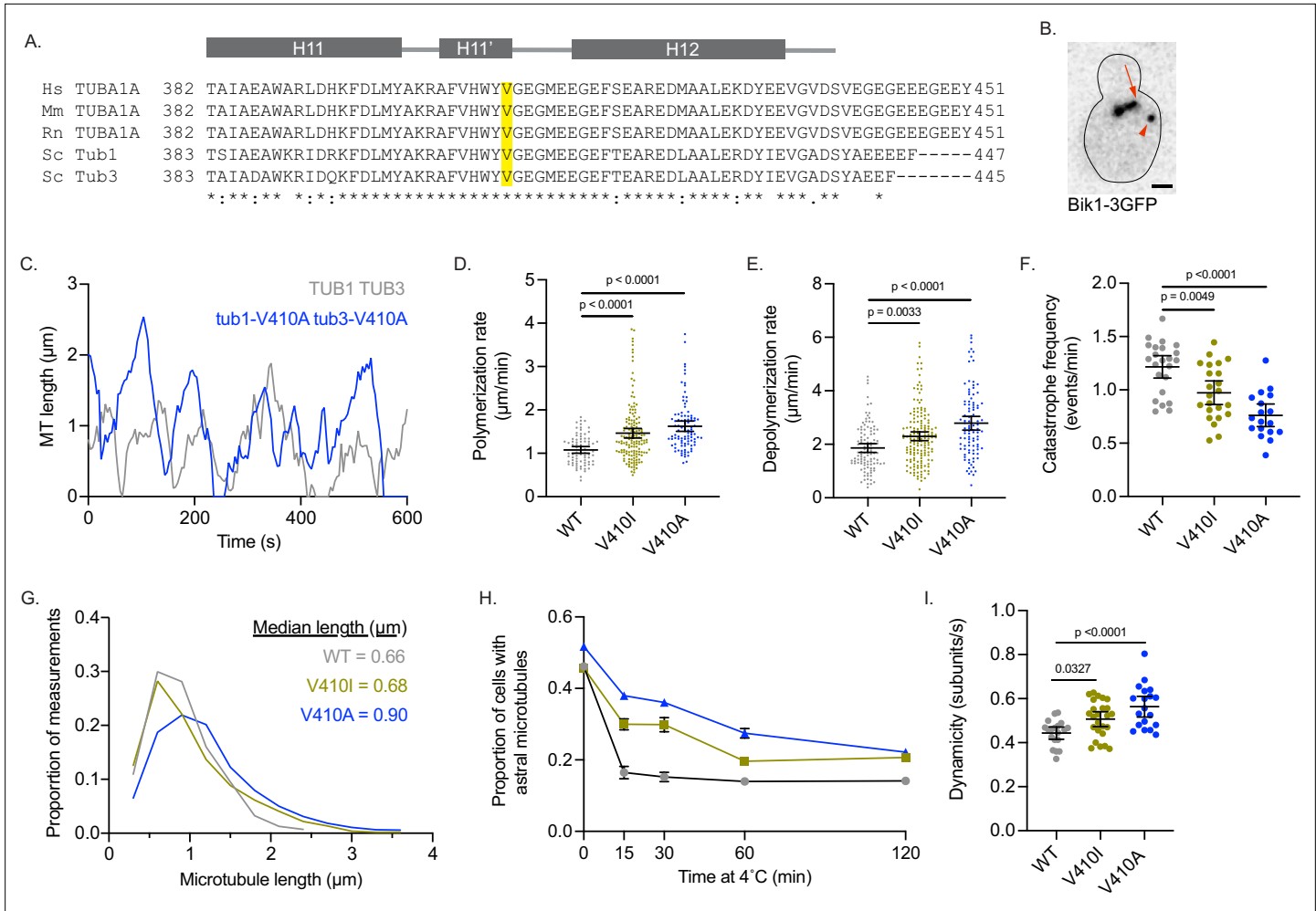

**Figure 4.** Modeling *TUBA1A*-V409I/A mutants in budding yeast reveals altered microtubule dynamics. (**A**) Sequence alignment of human, mouse, and rat TUBA1A and budding yeast α-tubulin Tub1 and Tub3. Valine 409 is highlighted in yellow and resides in a highly conserved helix known as H11'. (**B**) Representative image of budding yeast cell with microtubule plus end binding protein Bik1 labeled with 3GFP. Scale bar = 1 μm. (**C**) Life plot of the dynamics of individual microtubules from wild-type (WT) (gray) and V410A (blue) cells showing the change in length over time. (**D**) Polymerization rates of astral microtubules. Images were obtained from three independent experiments; n=19 cells for WT, n=25 for V410I, n=19 for V409A. Each dot represents a single polymerization event. Bars are mean ± 95% confidence interval. Statistical analysis was done using a one-way ANOVA followed by a Tukey test to correct for multiple comparison tests. All statistics with p≤0.05 are indicated on graph. (**E**) Depolymerization rates of astral microtubules. Each dot represents a single depolymerization event. (**F**) Catastrophe frequency for astral microtubules expressed as the number of catastrophe events that occur per minute. Each dot represents the average catastrophe frequency of a single cell. (**G**) Histogram of all astral microtubule lengths from time lapse imaging of WT, V410I, and V410A cells. (**H**) Proportion of cells with astral microtubules after indicated time at 4°C. Symbols represent average of three independent experiments; for WT n=581 cells at 0 min, n=385 at 15 min, n=362 at 30 min, n=365 at 60 min, n=283 at 120 min; for V409I n=574 cells at 0 min, n=502 at 15 min, n=235 at 30 min, n=383 at 60 min, n=484 at 120 min; for V409A n=677 cells at 0 min, n=401 at 15 min, n=257 at 30 min, n=247 at 60 min, n=238 at 120 min. (**I**) Dynamicity of astral microtubules calculated as the total change in length divided by the total change in time. Each dot represents the calculated dynamicity value of a single cell. Bars are mean ± 95% confidence interval. Statistical analysis was done using a one-way ANOVA followed by a Tukey test to correct for multiple comparison tests. All statistics with p≤0.05 are indicated on graph.

The online version of this article includes the following source data and figure supplement(s) for figure 4:

**Source data 1.** Quantification of microtubule dynamcis in yeast by Bik1-3GFP.

**Source data 2.** Proportion of cells with astral microtubules at cold temperature.

**Figure supplement 1.** Fitness assay for V410I/A yeast mutants.

**Figure supplement 1—source data 1.** Doubling time of yeast with WT or V410I/A tubulin.

in budding yeast indicate that the α-tubulin-V410I/A mutations are sufficient to alter microtubule dynamics in this system.

## tub1/tub3-V410I/A microtubules have decreased localization of and affinity for XMAP215/Stu2

Microtubule dynamics in cells are the product of both intrinsic tubulin activity and regulation by extrinsic MAPs. Therefore, we sought to determine how tub1/tub3-V410I/A affect extrinsic and/or intrinsic modes of regulation to alter microtubule dynamics. The α-tubulin V409 (human) or analogous V410 (yeast) residue resides on the external surface of the tubulin heterodimer near the binding sites of a wide variety of MAPs (Löwe et al., 2001). In particular, one structural analysis highlights the α-tubulin V410 residue as a potential interactor with the TOG2 domain of Crescerin1 (Das et al., 2015). The Crescerin1 TOG2 domain has a similar structure to the TOG1 domain of Stu2, the yeast homologue of XMAP215. Additionally, the V410 residue appears to reside at the interface between α-tubulin and the TOG1 domain of Stu2 (Figure 5A; Ayaz et al., 2012). As Stu2/XMAP215 is the major microtubule polymerase in cells, we predicted that the increase in microtubule polymerization observed in tub1/tub3-V410I/A cells could be a result of increased Stu2 activity on the mutant microtubules. To test this, we first used Stu2-3GFP to measure Stu2 localization at astral microtubule plus ends in cells where all the α-tubulin is either WT, V410I, or V410A mutant (Figure 5B). We find that tub1/tub3-V410A mutant microtubules have significantly decreased Stu2-3GFP fluorescence intensity at the plus ends compared to WT, while tub1/tub3-V410I microtubules have an intermediate phenotype between the two (3203±333, 1860±165, 1365±143 AU for WT, V410I, and V410A, respectively; Figure 5C). Thus, despite having increased microtubule polymerization rates, the tub1/tub3-V410I/A mutant microtubules have less Stu2 at microtubule plus ends.

Stu2 comprises two TOG domains (TOG1 and TOG2) that bind specifically to free tubulin heterodimer and are necessary for concentrating tubulin at microtubule plus ends to promote microtubule polymerization. Disrupting the interaction between the Stu2 TOG2 domain and tubulin heterodimers decreases the localization of Stu2 to microtubule plus ends (Geyer et al., 2018). We therefore predicted that the decreased localization of Stu2 at tub1/tub3-V410I/A microtubule plus ends may be a result of lowered affinity between the mutant tubulins and Stu2 TOG domains. To test this prediction, we purified tubulin heterodimer from budding yeast in which the purified α-tubulin was either Tub1-WT, -V410I, or -V410A. Additionally, we purified the two TOG domains in Stu2 fused to a GST tag (referred to as GST-TOG1/2). Using a low concentration of either WT or V410I/A mutant tubulin in the presence of increasing concentrations of GST-TOG1/2, we find that V410I, and more significantly V410A, mutant tubulin has decreased affinity for GST-TOG1/2 as compared to WT (Figure 5D and Figure 5—figure supplement 1A). For reference we used previously published $K_D$ values for WT yeast tubulin affinity for either TOG1 or TOG2 to determine an expected binding curve (Geyer et al., 2015). These results indicate that the decrease in Stu2 localization we see at the tub1/tub3-V410I/A microtubule plus ends is a consequence of the decreased affinity of Stu2 TOG1/2 domains for the mutant tubulin.

STU2 is an essential gene in budding yeast, presumably because of its important role in regulating microtubule dynamics (Wang and Huffaker, 1997). Based on the increased microtubule polymerization rates observed for V410I/A microtubules, and the decreased affinity between V410I/A and Stu2 TOG domains, we predicted that tub1/tub3-V410I/A microtubules would be resistant to the depletion of Stu2. To test this, we constructed tub1-V410I or -V410A yeast strains where we could conditionally deplete Stu2 (Kosco et al., 2001). Upon addition of copper to the media of these cells, synthesis of STU2 mRNA is repressed and Stu2 protein is degraded. We find that after 1.5 hr in 500 µM copper sulfate (CuSO₄), Stu2-HA is almost undetectable via western blot (Figure 5D and Figure 5—figure supplement 1B). Under these conditions, the proportion of WT cells with visible astral microtubules decreases from 53% to 34% (0.526±0.036, 0.345±0.27 at 0 and 1.5 hr, respectively; Figure 5E–F, Figure 5—figure supplement 1C). In contrast, tub1-V410I and tub1-V410A cells have no significant decrease in proportion of cells that have astral microtubules after 1.5 hr of Stu2 depletion (V410I: 0.537±0.023, 0.516±0.022; V410A: 0.524±0.036, 0.588±0.011 at 0 and 1.5 hr, respectively; Figure 5E–F, Figure 5—figure supplement 1C). These data indicate that V410I and V410A microtubules persist in the absence of Stu2, and therefore the mutant tubulins may not require Stu2 polymerase activity to the same extent as WT.

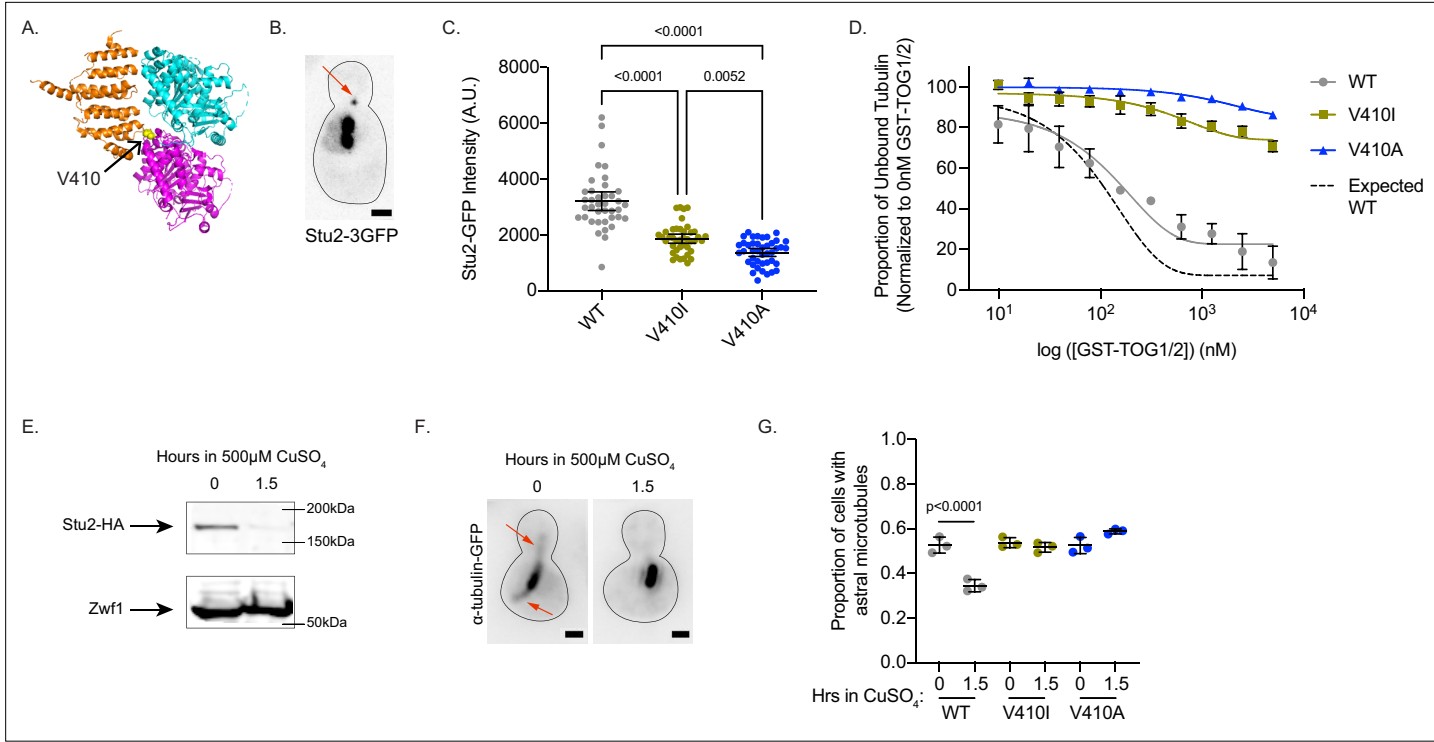

**Figure 5.** V410I/A mutants subvert Stu2 regulation. (**A**) Structure of yeast tubulin heterodimer with Stu2 TOG1 modified from PDB: 4FFB (*Ayaz et al., 2012*). Tub1-V409 residue displayed as yellow ball structure and labeled in zoomed in portion. (**B**) Representative image of budding yeast cell expressing Stu2-3GFP. Red arrow indicates Stu2 at microtubule plus end. Scale bar = 1 μm. (**C**) Quantification of Stu2-3GFP fluorescence intensity at microtubule plus ends. Images were acquired from three independent experiments; n=39 cells for wild-type (WT), n=42 for V410I, n=42 for V409A. Each dot represents the quantification from a single cell. Bars are mean ± 95% confidence interval. Statistical analysis was done using a one-way ANOVA followed by a Tukey test to correct for multiple comparison tests. All statistics with p≤0.05 are indicated on graph. (**D**) Proportion of unbound tubulin in solution in the presence of increasing concentrations of GST-TOG1/2. Data was analyzed from the α-tubulin signal on supernatant samples run on western blots. Each condition was normalized to the constant amount of tubulin added to each sample in the absence of GST-TOG1/2. Dots represent averages from three separate experiments. Bars are standard error of the mean. (**E**) Western blot of protein lysate from cells induced with 500 μM CuSO₄ for 0 and 1.5 hr. Blots were probed for HA and Zwf1. (**F**) Example images of WT cells induced with 500 μM CuSO₄ for 0 and 1.5 hr stained for α-tubulin. Red arrows indicate astral microtubules. Scale bar = 1 μm. (**G**) Quantification of proportion of cells imaged that have at least one astral microtubule. Images were acquired from three independent experiments; for WT n=1389 cells at 0 hr, n=1,525 at 1.5 hr; for V409I n=940 cells at 0 hr, n=2221 at 1.5 hr; for V409A n=1483 at 0 hr, n=1677 at 1.5 hr. Each dot represents the average proportion of cells from the three separate experiments. Bars are mean ± standard error of the mean. Statistical analysis was done using a one-way ANOVA followed by a Tukey test to correct for multiple comparison tests. All statistics with p≤0.05 are indicated on graph.

The online version of this article includes the following source data and figure supplement(s) for figure 5:

**Source data 1.** Fluorescence intensity of Stu2-3GFP at microtubule plus ends.

**Source data 2.** Proportion of unbound tubulin in the presence of TOG domains.

**Source data 3.** Proportion of cells with astral microtubules following Stu2 depletion.

**Source data 4.** Western blot of HA-tagged Stu2 in the absence and prescence of 500μM copper sulfate.

**Source data 5.** Labeled western blot of HA-tagged Stu2 in the absence and prescence of 500μM copper sulfate.

**Figure supplement 1.** V410I/A mutants affect Stu2 interactions and regulation.

**Figure supplement 1—source data 1.** Labeled western blot of WT alpha tubulin in the presence of increasing concentrations of GST-TOG1/2.

**Figure supplement 1—source data 2.** Western blot of WT alpha tubulin in the presence of increasing concentrations of GST-TOG1/2.

**Figure supplement 1—source data 3.** Western blot of V410I alpha tubulin in the presence of increasing concentrations of GST-TOG1/2.

**Figure supplement 1—source data 4.** Labeled western blot of V410I alpha tubulin in the presence of increasing concentrations of GST-TOG1/2.

**Figure supplement 1—source data 5.** Western blot of V410A alpha tubulin in the presence of increasing concentrations of GST-TOG1/2.

**Figure supplement 1—source data 6.** Labeled western blot of V410A alpha tubulin in the presence of increasing concentrations of GST-TOG1/2.

**Figure supplement 1—source data 7.** Western blot of time course of HA-tagged Stu2 in the prescence of 500μM copper sulfate.

*Figure 5 continued on next page*

*Figure 5 continued*

**Figure supplement 1—source data 8.** Labeled western blot of time course of HA-tagged Stu2 in the presence of 500µM copper sulfate.

**Figure supplement 1—source data 9.** Western blot of Zwf1 loading control for time course of HA-tagged Stu2 in the presence of 500µM copper sulfate.

**Figure supplement 1—source data 10.** Labeled western blot of Zwf1 loading control for time course of HA-tagged Stu2 in the presence of 500µM copper sulfate.

## Purified *tub1*-V410A has increased microtubule polymerization rates in vitro

Stu2 TOG domains preferentially bind to the kinked conformation of tubulin and have low affinity for straight heterodimers (*Ayaz et al., 2014*; *Ayaz et al., 2012*). Based on our in vitro binding assays, we hypothesized that α-tubulin V410I/A heterodimers may adopt a straighter state. The V410 residue resides in helix 11' of α-tubulin, which is located at the hinge point between α- and β-tubulin (*Löwe et al., 2001*), and makes it a prime candidate for affecting the conformational states of the heterodimer. We predicted that a straighter heterodimer would increase microtubule polymerization rates as a straight heterodimer is more compatible with forming the microtubule lattice. Therefore, we reasoned that if the changes observed in V410I/A microtubule dynamics are a result of a straightened heterodimer conformation, the mutants should have intrinsically faster polymerization rates in vitro.

To test this, we used interference reflection microscopy (IRM) to measure the intrinsic polymerization activity of our purified yeast Tub1-WT, -V410I, and -V410A heterodimers in an in vitro dynamics assay. We used GMPCPP-stabilized, rhodamine-labeled porcine microtubule 'seeds' attached to the surface of the coverslip via anti-rhodamine antibodies to nucleate the assembly of microtubules from the purified tubulin in the reaction (*Figure 6A*). We used three concentrations of purified tubulin, between 0.5 and 0.9 µM, to measure microtubule dynamics in vitro. Within this concentration range, we find that soluble tubulin assembles from the stabilized seeds and forms dynamic microtubules. If tubulin concentration is too high, the tubulin will spontaneously nucleate away from the stabilized seeds (to form microtubules, oligomers, aggregates, etc.) and will not be visible on the microscope.

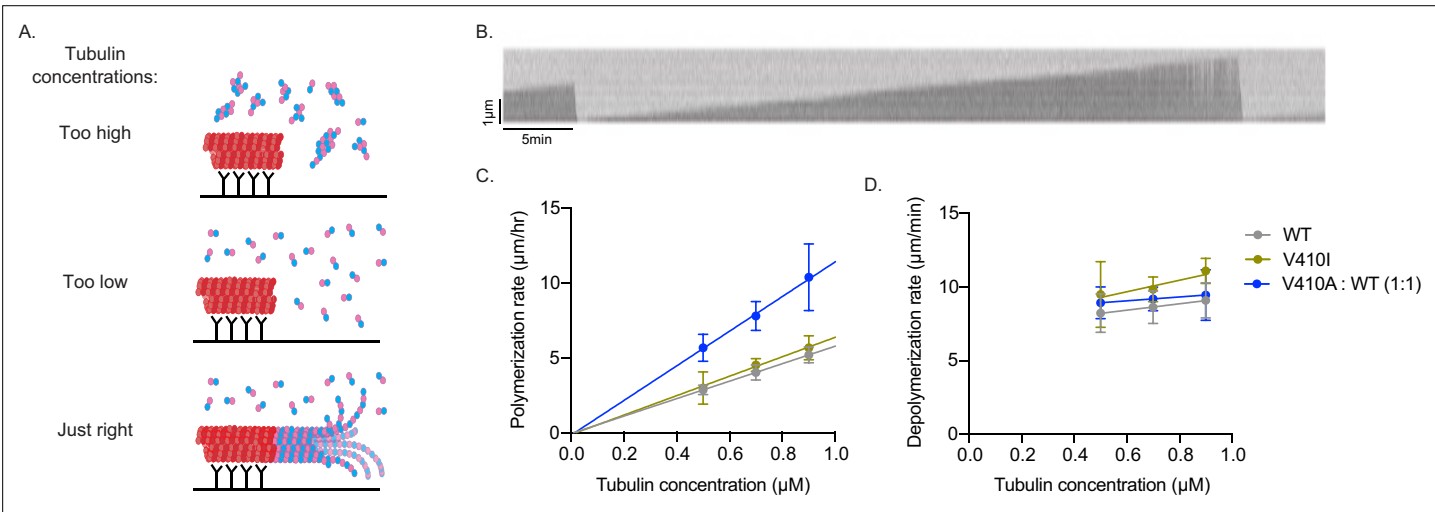

**Figure 6.** V410A has intrinsically faster microtubule polymerization rates in vitro. (**A**) Graphic depicting aggregates and non-seeded nucleation when tubulin concentrations are too high, no nucleation when concentrations are too low, and dynamic microtubules when concentrations are at an appropriate level. (**B**) Example kymograph of dynamic microtubule from purified budding yeast tubulin measured using interference reflection microscopy. X and Y scale bars = 5 min and 1 µm, respectively. (**C**) Microtubule polymerization rates at 0.5, 0.7, and 0.9 µM of tubulin. Each data point represents the mean of three independent experiments; for wild-type (WT) n=15 microtubules at 0.5 µM, n=23 at 0.7 µM, n=18 at 0.9 µM; for V410I n=12 microtubules at 0.5 µM, n=14 at 0.7 µM, n=12 at 0.9 µM; for V410A n=16 microtubules at 0.5 µM, n=20 at 0.7 µM, n=10 at 0.9 µM. Bars are mean ± 95% confidence interval. (**D**) Microtubule depolymerization rates.

The online version of this article includes the following source data for figure 6:

**Source data 1.** In vitro microtubule dynamics.

**Source data 2.** Microtubule dynamics in vitro.

If the tubulin concentration is too low, the tubulin will not sustain assembly from the seeds. In our experimental setup, we are unable to observe microtubule dynamics below 0.5 µM tubulin. For each concentration, we measured the change in microtubule length over time of WT or V410I/A mutant tubulin in a completely purified system using unlabeled tubulin (*Figure 6B*, *Figure 6—source data 2*).

In contrast to our results in budding yeast cells, we find that purified tub1-V410I microtubules do not have significantly increased polymerization rates as compared to WT at any of the three concentrations tested (*Figure 6C*). However, with purified tub1-V410A tubulin, we were unable to observe microtubule dynamics at these three concentrations, nor did we observe any at higher concentrations up to 1.5 µM or lower concentrations down to 0.1 µM (data not shown). Based on our previous data, we predict that tub1-V410A tubulin exhibits increased assembly activity and would be more dynamic at lower concentrations. If tub1-V410A tubulin exhibits higher assembly activity than WT tubulin, even the low end of our usable concentration range may be too high for this mutant, and we would be unable to visualize microtubule dynamics because the mutant may readily assemble away from seeds (similar to top panel of *Figure 6A*). To circumvent these issues while still testing the intrinsic capabilities of tub1-V410A, we mixed WT and tub1-V410A tubulin in a one-to-one ratio, for a total tubulin concentration of 0.5, 0.7, or 0.9 µM. At this WT:V410A one-to-one ratio, we find an increase in microtubule polymerization rates at each concentration tested (*Figure 6C*). Since the amount of either WT or V410A tubulin that is present in each of these one-to-one mixtures (i.e., 0.25, 0.35, or 0.45 µM) is not sufficient to support microtubule assembly on its own, we conclude that the increased microtubule polymerization is a synergistic effect of the blend of WT and V410A mutant tubulin.

The fitted lines of the data collected at each tubulin concentration provide us with important information about the intrinsic properties of the WT and mutant microtubules formed in vitro. The slopes of the fitted lines represent the concentration-dependent polymerization rate, and we find that WT/V410A has an ~2-fold increased rate as compared to WT and V410I (11.6 compared to 5.8 and 6.5 µm/hr/µM). The x- and y-intercepts represent the critical concentration and apparent off rate constants, respectively (x-intercepts=0.01, 0.02, and 0.02 µM; y-intercepts=–0.07, –0.13, and –0.18 µm/hr for WT, V410I, and WT/V410A, respectively). In contrast to the increase observed in microtubule polymerization rates, the microtubule depolymerization rates were not different between WT and either the V410I or V410A mutants (*Figure 6D*). These data indicate that *tub1*-V410A has significantly increased intrinsic microtubule polymerization as compared to WT. Together, our data explain the intrinsic mechanism by which the V409/V410A mutant has the most severe phenotypes across scales, while V409/V410I has more mild effects.

## Discussion

The missense, heterozygous mutations identified in human tubulin genes and associated with brain malformations are numerous and varied (*Bahi-Buisson et al., 2014*; *Fallet-Bianco et al., 2014*). To date, we do not have a good understanding of how different mutations in tubulin genes have such a wide spectrum of detrimental effects on brain development. Investigating the molecular impact of different tubulinopathy mutations and how these mutants affect greater tissue level development is critical to advancing our understanding of the role of the microtubule cytoskeleton during neurodevelopment and disease. In this study, we aim to determine the mechanism of two different substitutions at the valine 409 residue of *TUBA1A* that are associated with different severities of brain malformations. We find that ectopically expressed *TUBA1A*-V409I/A mutants assemble into microtubules in neuron cultures and dominantly disrupt radial neuron migration, and most significantly in cells expressing *TUBA1A*-V409A (*Figure 1*). Our work indicates that the mutation associated with the most severe brain phenotype, *TUBA1A*-V409A, also has the most severe impact on neurite branching and microtubule dynamics, while the milder -V409I mutant has intermediate phenotypes. These findings provide evidence that the impact of tubulinopathy mutations scale up from altered microtubule polymerization activity and loss of regulation by MAPs to cellular and tissue defects.

As neurons migrate from the ventricular zone to the cortical plate, they must transition through different polarization states (*Nadarajah et al., 2001*; *Noctor et al., 2004*; *Tabata and Nakajima, 2003*). The overexpression or depletion of a wide variety of signaling molecules and cytoskeletal proteins delay or inhibit neuron morphology transitions that impair subsequent migration (reviewed in *Cooper, 2014*). Several cases of brain malformations similar to those investigated in this study have been linked to mutations that disrupt radial migration by inhibiting the multipolar-to-bipolar transition

(*Ohshima et al., 2007*). While there are numerous factors that are required for proper neuron migration, our data support a hypothesis that *TUBA1A*-V409I/A mutants disrupt and/or delay proper morphology transitions. At the cellular level, we find that neurons ectopically expressing *TUBA1A*-V409I and -V409A disrupt neuron morphogenesis by promoting excessive branch stabilization (*Figures 1F and 2A–E*). We find that *TUBA1A*-WT and *TUBA1A*-V409I/A expressing neurons extend and retract neurite branches at similar rates, however the mutant-expressing cells have neurites that retract less often and spend more time in a paused state (*Figure 2D–E*). In the context of a developing cortex, neurons need to efficiently extend and retract neurites to sample the surrounding space, respond to developmental cues, and ultimately polarize to form a bipolar neuron. Our data suggests that the *TUBA1A*-V409I/A mutant-expressing cells have a more elaborate yet static morphology, with neurites that spend less time sampling space as compared to WT neurons. Our findings present the question of whether *TUBA1A*-V409I/A mutants completely fail to migrate to the cortical plate, or whether migration is merely delayed. Future studies addressing this question, potentially by monitoring the migration of neurons at later developmental time points or via in vitro slice cultures, will provide insight into whether neuronal migration is completely impaired or simply delayed. To the best of our knowledge, there are no tubulinopathy mutations to date that disrupt neuron migration via impaired neuron morphology transitions. Therefore, future work will be required to determine whether this is a common mechanism for particular types of brain malformations.

Our findings support the idea that microtubule stability plays a determining role in neuron branching and morphogenesis. Previous studies have identified that taxol-treated neurons have an increase in neurite extension, axon formation, and branching, while nocodazole treatments result in diminished neurite extension and increased retraction (*He et al., 2002*; *Witte et al., 2008*). The altered expression of, or mutations in, MAPs also alter neuron morphogenesis (reviewed in *Lasser et al., 2018*). For example, the MAP SSNA1 promotes axonal branching by mediating microtubule nucleation and branching, further highlighting the role of microtubule dynamics in neuron morphogenesis (*Basnet et al., 2018*). Neurons must tightly control both microtubule turnover and nucleation to control neurite branching (*Basnet et al., 2018*; *Witte et al., 2008*). Both taxol treatment and SSNA1 overexpression result in increased neurite branching, yet each drives this phenotype via separate mechanisms; Taxol suppresses microtubule turnover, whereas overexpression of SSNA1 induces microtubule nucleation and branching. We find that DIV2 neurons ectopically expressing *TUBA1A*-V409I or -V409A are resistant to cold-induced neurite loss and have increased levels of microtubule acetylation (*Figures 2F–H, and 3C*). Our analysis of microtubule dynamics in neurons reveals a similar number of polymerizing microtubules, but faster polymerization rates when V409I and V409A is expressed, compared to WT (*Figure 3F and G*). We also find that the duration of polymerization events is shorter when V409I and V409A is expressed, compared to WT (*Figure 3H*). Whether these shorter polymerization events are followed by transitions to depolymerization (i.e., catastrophe) or to a stable, non-polymerizing state is unclear. Our data suggest that the V409 mutants do not promote increased nucleation as compared to WT, but rather V409I, and more strongly V409A, increase the content of stable microtubule polymer in neurons. This may be due in part to increased polymerization rates; however, we cannot rule out changes in transition frequencies or depolymerization rate, neither of which are accessible in our dynamics data.

Our results from modeling V409 mutants in budding yeast further reveal the mechanistic origins of the highly stable microtubules in neurons. Similar to our results in neurons we find that, compared to WT, *tub1/tub3*-V410I/A microtubules have faster polymerization rates (*Figure 4D*). In yeast we are able to analyze additional microtubule dynamics parameters, and we find that *tub1/tub3*-V410I/A microtubules also have increased depolymerization rates, decreased catastrophe frequencies, and increased dynamicity (*Figure 4E, F,I*). Dynamicity is calculated as the total change in microtubule length (taking into account states of growth and shrinkage) divided by the change in time (*Jordan et al., 1993*). It is interesting that despite the *tub1/tub3*-V410I/A microtubules undergoing catastrophe less often than WT, they have increased dynamicity values. This suggests that the increase in dynamicity is driven by increased polymerization and depolymerization rates, as opposed to frequent transitions between the states of polymerizing and depolymerizing. Accordingly, V410I/A microtubules reach longer lengths and are longer lived. Assuming these dynamic parameters hold in neurons (as we find the polymerization rates do), longer microtubules that catastrophe less often could also contribute to the increase in microtubule acetylation that we observe in neurons expressing *TUBA1A*-V409I/A. Our data support

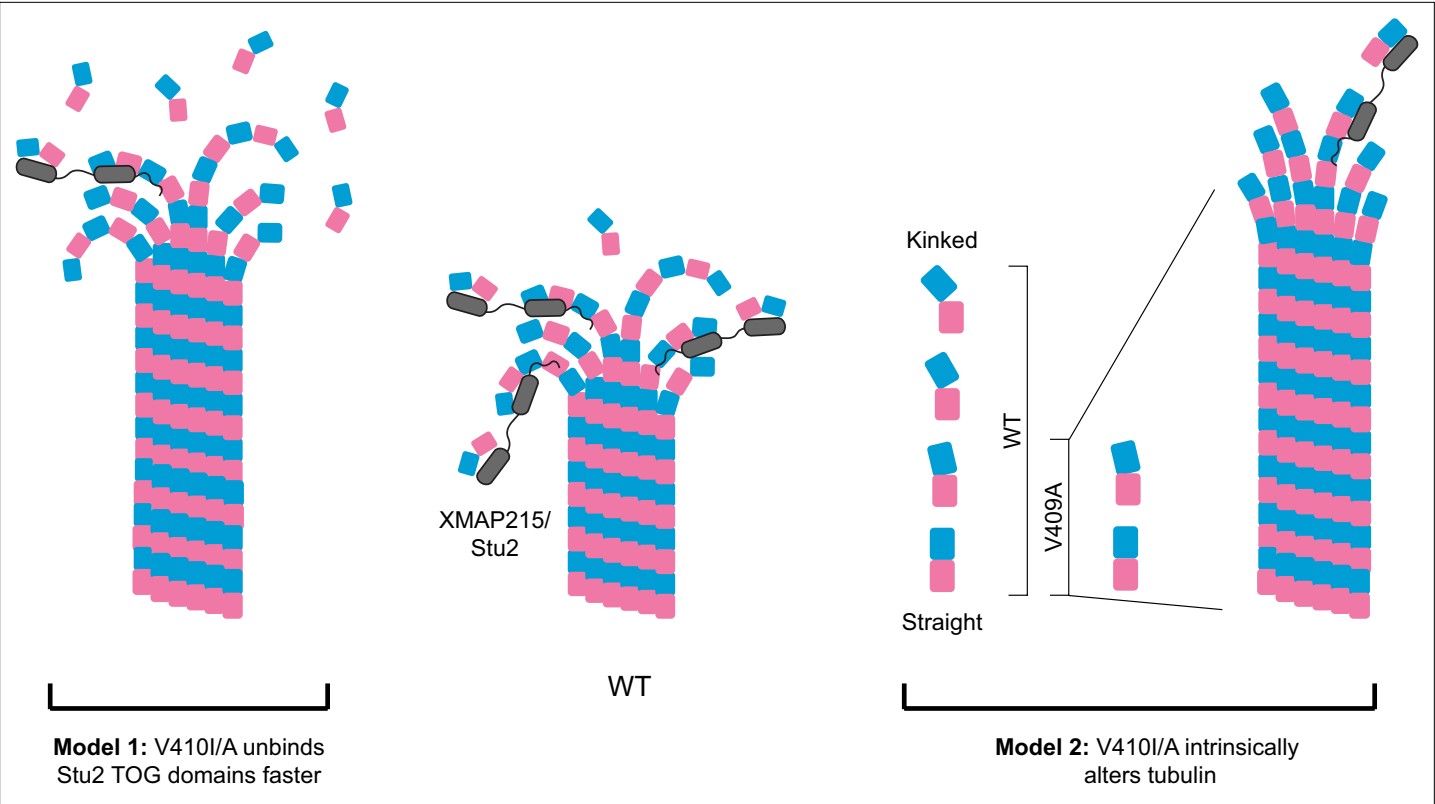

**Figure 7.** Potential models for the effect of α-tubulin V409I/A on XMAP215/Stu2 and microtubule polymerization. Wild-type (WT) tubulin undergoes a series of conformational states as it transitions from a curved free heterodimer into the straight microtubule lattice, which is essential for tubulin binding to a variety of microtubule-associated proteins (MAPs), such as XMAP215/Stu2 (center). Our data suggest that V409I, and more severely V409A, have decreased XMAP215/Stu2 activity at microtubule plus ends, yet have faster microtubule polymerization rates. **Model 1:** V410I/A tubulin unbinds from TOG (tumor overexpressed gene) domains faster, thus increasing the concentration of free tubulin to promote faster microtubule polymerization. However, if the faster microtubule polymerization rates were solely driven by a decrease in XMAP215/Stu2 and/or free tubulin concentration, we would not expect to see an increase in intrinsic polymerization rates in V409A as compared to WT, as we do in our reconstitution experiments. Therefore, we favor the second model. **Model 2:** the V410I/A tubulin heterodimer is intrinsically altered in a way that disrupts normal XMAP215/Stu2 regulation and drives faster microtubule polymerization. One way in which tubulin could be intrinsically changed is through adopting a straighter, lattice-favoring conformation.

a model in which the V409I/A mutants promote excessive neurite branching similar to taxol-treated neurons, but alter microtubule dynamics in a way that is distinct from the mechanisms of taxol.

Microtubule dynamics in cells are reliant on both extrinsic factors that interact with microtubules and intrinsic factors conferred by the tubulin heterodimer (*Bodakuntla et al., 2019*; *Borys et al., 2020*; *Goodson and Jonasson, 2018*; *Manka and Moores, 2018*; *Mitchison and Kirschner, 1984*). One way in which tubulin heterodimers intrinsically control microtubule dynamics is by adopting a series of conformations as microtubules polymerize and depolymerize (*Chrétien et al., 1995*; *Mandelkow et al., 1991*). Free heterodimer exists in a kinked conformation, which straightens out as the heterodimer is assembled into the microtubule lattice (*Buey et al., 2006*; *Jánosi et al., 1998*; *Nawrotek et al., 2011*; *Nogales and Wang, 2006*; *Ravelli et al., 2004*; *Rice et al., 2008*). XMAP215/Stu2 is thought to play an important role in driving microtubule polymerization by binding free tubulin near microtubule plus ends and facilitating the transition from the kinked, free heterodimer to the straight heterodimer in the microtubule lattice (*Figure 7*). We find that V410I/A tubulins have weaker affinity for Stu2 TOG1/TOG2 domains in our in vitro binding assays, and that V410I microtubules, and more significantly -V410A microtubules, have less Stu2 at their plus ends in yeast cells (*Figure 5C and D*). Our finding that V409I, and more so V409A, have decreased XMAP215/Stu2 activity and faster microtubule polymerization rates seem paradoxical.

*Figure 7* depicts two models that could reconcile weak TOG binding with faster polymerization: (1) V410I/A mutant tubulin unbinds faster from TOG domains and increases the concentration of free

tubulin available to polymerize into microtubules, and (2) V410I/A alters the intrinsic tubulin conformation to constitutively adopt a straighter, lattice-favoring state. These two models are not mutually exclusive as it is common for MAPs to influence tubulin conformation states and vice versa. However, these two models can be distinguished by their dependence on TOG binding. The first model posits that V410I/A tubulin exhibits faster polymerization than WT tubulin because WT tubulin visits a slow step in the microtubule assembly pathway when it is bound to the TOG domains of XMAP215/Stu2. In the second model, V410I/A tubulin exhibits faster polymerization regardless of whether XMAP215/Stu2 is present. Our finding that V410A increases the rate of tubulin polymerization in the absence of extrinsic MAPs favors the second model and is consistent with an effect on tubulin's intrinsic conformation (*Figure 6C*). Additionally, when Stu2 is conditionally depleted in cells, *tub1/tub3*-V410I/A microtubules are maintained while WT microtubules are lost (*Figure 5G*). Further experiments are needed to directly test whether V410 mutants limit the range of heterodimer conformations to adopt an intrinsically straighter state. At the cellular level, this model predicts that limiting tubulin conformational states could diminish regulation that is typically conferred by Stu2.

Tight control over microtubule dynamics is essential for cells, and particularly in neuron morphogenesis and migration as the cytoskeletal must nimbly respond to cues throughout the migratory process (*Kapitein and Hoogenraad, 2015*). XMAP215/Stu2 works in cells as an extrinsic regulator to finely tune microtubule dynamics (*Brouhard et al., 2008*; *Hahn et al., 2021*; *Lee et al., 2001*; *van der Vaart et al., 2011*; *Vasquez et al., 1994*; *Zanic et al., 2013*). XMAP215's polymerase activity is required for axon outgrowth and microtubule bundling, and in the growth cone XMAP215 plays a role in linking microtubules and F-actin (*Hahn et al., 2021*; *Lowery et al., 2013*; *Slater et al., 2019*). When XMAP215 is depleted in neurons, the number of polymerizing microtubules and the rate of polymerization decrease, and axons are shorter due to slower axon growth rates (*Hahn et al., 2021*; *Lowery et al., 2013*). As most of the role of XMAP215 in neurons has been explored primarily in the axon and growth cone, future studies should focus on the role XMAP215 plays in neuron morphogenesis and migration, including in establishing neuron polarity and branching. Beyond XMAP215/Stu2, which preferentially binds to the kinked conformation of tubulin, there are numerous other MAPs that bind to a range of particular tubulin conformations (*Brouhard and Rice, 2018*). Therefore, we predict that the activity of multiple extrinsic regulators would be impacted by mutations that alter the conformational state of the tubulin heterodimer. How these regulatory mechanisms are impacted by tubulin conformational states and the subsequent effects of these changes on microtubule networks in cells will be a major focus of future studies.

Our results, particularly for the V410A mutant, highlight how tubulin mutants can create unexpected dissonance between extrinsic and intrinsic factors. Compared to WT tubulin, V410A mutant tubulin promotes faster polymerization rates in cells and in vitro (*Figures 3G, 4D and 6C*). It is interesting that in our in vitro system, we were unable to identify a concentration at which homogeneous V410A polymerizes from the GMPCPP-stabilized porcine tubulin seeds. However, blending V410A with WT tubulin in a one-to-one ratio supports the assembly of microtubules with significantly increased polymerization rates as compared to WT at the same concentrations of total tubulin. This suggests that V410A may be unable to polymerize from GMPCPP seeds on its own, but it drives fast polymerization when mixed with WT tubulin. Based on our data acquired from neuron cultures, and particularly from budding yeast in which all the α-tubulin in the cell is V410A, we know that this mutant assembles into microtubules that polymerize faster and are longer lived in cells. Why then is V410A able to assemble into microtubules when it is the only source of α-tubulin in the cell, but not when it is the only source of α-tubulin in vitro? Beyond V410A, how synergistic interactions between different tubulin heterodimers occur (such as different mutants or isotypes) is an important question to address both in regard to tubulinopathy disease as well as normal tubulin biology. In tubulinopathy patients that harbor heterozygous mutations, the mutant tubulin comprises at most 50% of either the α- or β-tubulin pool. This underscores the necessity of future work to (1) determine the relative protein constituency of different tubulin isotypes in different cell types and (2) use blending experiments in vitro to recapitulate these relative proportions. Additionally, this is an important consideration for furthering our understanding of how normal tubulin biology works in cells. Using mutants that affect particular intrinsic properties of tubulin (e.g., conformation, GTPase activity, etc.) in various blends with WT tubulin will provide insight into how these different properties work in trans to alter microtubule dynamics, and whether some mutants or isotypes might elicit outsized effects.

We propose a model in which the V409I/A mutants are intrinsically altered in a way that perturbs XMAP215/Stu2 regulation and ultimately drive faster microtubule dynamics that perturb neuron morphologies and migration (*Figure 7*). An isoleucine or alanine substitution at the valine 409 residue of *TUBA1A* seems relatively insignificant. However, these subtle changes lead to drastic consequences that scale from altered microtubule dynamics to an underdeveloped neocortex. Strikingly, *TUBA1A*-V409I presents an intermediate phenotype between WT and *TUBA1A*-V409A in each assay we tested, tracking with the observed brain malformations observed in these patients. To date, there are numerous missense, heterozygous mutations in human tubulin genes that are associated with brain malformations, yet the molecular mechanisms driving these tissue level defects remain unknown. Based off our work and that of others, there does not appear to be a consistent mechanism for how *TUBA1A* tubulinopathy mutations affect microtubule networks and neuron cell biology (*Aiken et al., 2019*; *Belvindrah et al., 2017*; *Gartz Hanson et al., 2016*; *Keays et al., 2007*). The well-established conformational dynamics of tubulin raise the possibility that mutations altering one region of the heterodimer could allosterically affect distant regions, and thus disrupt the normal conformational transitions that heterodimers undergo during assembly and disassembly. Therefore, it is crucial to use quality genetic models to further our understanding of the molecular mechanism of these patient-associated mutations. Additionally, our work highlights the need for more detailed studies into how different regions of the tubulin heterodimer impact tubulin conformational states, and how this affects extrinsic regulation that ultimately result in altered microtubule dynamics. Investigating how subtle alterations in specific regions of the heterodimer can have long ranging effects across microtubule networks will be a primary focus of future work.

# Materials and methods

## Key resources table

| Reagent type (species) or resource | Designation | Source or reference | Identifiers | Additional information |
|---|---|---|---|---|
| Gene (*Homo sapiens*) | *TUBA1A* | NCBI | Gene ID: 7846 | |
| Gene (*Saccharomyces cerevisiae*) | *TUB1* | *Saccharomyces* genome database | SGD:S000004550 | |
| Gene (*Saccharomyces cerevisiae*) | *TUB3* | *Saccharomyces* genome database | SGD:S000004593 | |
| Biological sample (*Mus musculus*) | In utero electroporation coronal brain slices | This paper | | Coronal slices of E18.5 mouse brains |
| Biological sample (*Rattus norvegicus*) | Primary cortical rat neurons | This paper | | Cortices obtain from P0-P2 rats |
| Recombinant DNA reagent | pCIG2-TUBA1A-6X-His | *Buscaglia et al., 2020a* | | Human *TUBA1A* plasmids transfected in mouse or rat neurons |
| Recombinant DNA reagent | pCIG2-TUBA1A-IRES-GFP-MACF43 | *Aiken et al., 2019* | | Plasmid to co-express human TUBA1A and GFP-MACF43 in rat neurons |
| Sequence-based reagent | TUBA1A-V409A F QC | This paper | PCR primer; p674 | ctttgttcactggtacgctggggaggggatg |
| Sequence-based reagent | TUBA1A-V409A R QC | This paper | PCR primer; p675 | catcccctccccagcgtaccagtgaacaaag |
| Sequence-based reagent | TUBA1A-V409I F QC | This paper | PCR primer; p986 | cctttgttcactggtacattggggaggggatgg |
| Sequence-based reagent | TUBA1A-V409I R QC | This paper | PCR primer; p987 | ccatcccctccccaatgtaccagtgaacaaagg |
| Sequence-based reagent | tub1-V410A F | This paper | PCR primer; p545 | CGTCCACTGGTATGCCGGTGAAGGTATG |
| Sequence-based reagent | tub1-V410A R | This paper | PCR primer; p546 | CATACCTTCACCGGCATACCAGTGGACG |
| Sequence-based reagent | tub1-V410I F | This paper | PCR primer; p1018 | cgtgctttcgtccactggtatatcggtgaaggt |
| Sequence-based reagent | tub1-V410I R | This paper | PCR primer; p1019 | accttcaccgatataccagtggacgaaagcacg |
| Commercial assay, kit | Amaxa rat nucleofector kit | Lonza | VPG-1003 | For transfecting primary cortical neurons |
| Antibody | Anti-acetylated tubulin (mouse monoclonal) | Sigma | T7451 | IF (1:1000) |
| Antibody | Anti-tyrosinated tubulin (rat monoclonal) | Sigma | MAB1864-I | IF (1:1000) |

*Continued on next page*

*Continued*

| Reagent type (species) or resource | Designation | Source or reference | Identifiers | Additional information |
|---|---|---|---|---|
| Antibody | Anti-6X-His (mouse monoclonal) | Invitrogen | 4A12E4 37–2900 | IF (1:500) |
| Antibody | Anti-beta II tubulin (rabbit monoclonal) | Abcam | ab179512 | IF (1:500) |
| Antibody | Anti-rabbit IgG AF568 (goat polyclonal) | Thermo | A-11011 | IF (1:200) |
| Antibody | Anti-rat IgG AF568 (goat polyclonal) | Thermo | A-11077 | IF (1:200) |
| Antibody | Anti-mouse IgG AF647 (goat polyclonal) | Thermo | A32728 | IF (1:200) |
| Other | DAPI fluoromount-G | Southern Biotech | 0100–20 | Used in *Figure 1C and D* |
| Recombinant DNA reagent | pGEX6p1-GST-STU2 1–590 | *Widlund et al., 2012* | | Expression of GST-fused Stu2 TOG1/2 domains |
| Recombinant DNA reagent | pRS426-GAL-Tub1-internal 6X-His | *Johnson et al., 2011* | | Overexpress α-tubulin in yeast |
| Recombinant DNA reagent | pRS424-GAL-Tub2 (untagged) | *Johnson et al., 2011* | | Overexpress β-tubulin in yeast |
| Strain, strain background (*Saccharomyces cerevisiae*) | Jel1 | *Lindsley and Wang, 1993* | | Protease deficient yeast strain used for expressing tubulin for purification |
| Strain, strain background (*Saccharomyces cerevisiae*) | YEF473 | *Bi and Pringle, 1996* | | Lab yeast strain used for all genetic and cell biology experiments |
| Strain, strain background (*Saccharomyces cerevisiae*) | Stu2 depletion strains | *Kosco et al., 2001* | In this paper, y4835, y4836 | Depletion of Stu2 from strain upon addition of 500 µM CuSO$_4$ |
| Strain, strain background (*Escherichia coli*) | BL21 | Invitrogen | D1306 | Competent bacterial cells |
| Software, algorithm | U-Track | *Applegate et al., 2011* | | Tracking microtubule plus ends in *Figure 3E–H* |

## Animals

Animal research was performed following regulations approved by the Institutional Animal Care and Use Committee at the University of Colorado School of Medicine. All mouse tissue experiments described below were from *C57BL/6J* WT pregnant mice (Jackson Laboratories Strain #000664). All dissociated neuron cultures described below were prepared from Sprague-Dawley rats. Dissociated cultures were obtained from both male and female pups.

## TUBA1A plasmid vectors

Plasmids used in this study are listed in *Appendix 1—table 1*. The plasmids were constructed using the methods described in *Aiken et al., 2019*. Briefly, the coding region of the human *TUBA1A* gene was cloned into the multiple cloning site of the pCIG2 plasmid that expresses cytoplasmic GFP from an internal ribosome entry site (IRES) (*Hand et al., 2005*). QuikChange Lightning Site-Directed Mutagenesis Kit (Agilent Technologies, Santa Clara, CA) was used to introduce either the p.V409A or p.V409I substitution into TUBA1A. All plasmids have the 6X-His tag inserted between residues I42 and G43. Mutants were confirmed by sequencing. Oligos used for plasmid constructions are listed in *Appendix 1—table 3*.

## In utero electroporation

In utero electroporations were performed on E14.5 embryos of *C57BL/6J* WT pregnant mice. Plasmid DNA was endotoxin-free and prepared with 0.1% fast green dye and had a final concentration of 1 µg/ µl in TE buffer (10 mM Tris base and 1 mM EDTA, pH 8.0). Plasmid DNA was injected into the lateral ventricles of the exposed embryos and electroporated with 5 pulses at 50 V for 100 ms pulses each, separated by 950 ms. All embryos developed to E18.5, then brains were dissected and fixed overnight at 4°C in 4% paraformaldehyde (PFA). A vibrating microtome (VT1200S; Leica, Buffalo Grove, IL) was used for 50 µm thick coronal sections. Sections were mounted on glass slides, then imaged

on a spinning disk confocal microscope with a 20× objective. Analysis includes at least three animals, both male and female, per condition. High-resolution, zoomed in images of ventricular/subventricular and intermediate zones were taken on a Stellaris 5 confocal microscope with a 63× oil objective and 5 HydS tunable detectors (Leica).

## Primary cortical neuron cultures

The frontal cortex was dissected from postnatal days 0–2 male and female neonatal Sprague-Dawley rats. Cortices were digested at room temperature for 1 hr using papain solution composed of 20 units/ml papain (Worthington Biochem, LS003126), 1.5 mM $CaCl_2$ (Sigma-Aldrich, 223506), 0.5 mM EDTA (Fisher Scientific, BP118-500), 1 mM NaOH (Fisher Scientific, BP359-500), and 0.2 mg/ml cysteine (Sigma-Aldrich, C6852) in 10 ml saline. Digested cortices were washed six times with Minimal Essential Media (MEM) (Life Technologies, Carlsbad, CA, 11090-081). Cortices were dissociated with a wide-bore fire-polished pipette, followed by dissociation with a narrow-bore polished pipette. pCIG2-TUBA1A-6X-His or pCIG2-TUBA1A-IRES-GFP-MACF43 (either WT, V409I, or V409A) plasmids (4 µg) were introduced to $4 \times 10^6$ dissociated neurons using Amaxa rat nucleofector kit (Lonza Bioscience, VPG-1003). Amaxa-nucleofected neurons plated at 500,000 cells per 35 mm poly-D-lysine-coated glass bottom dish (WillCo Wells, HBST-3522) in MEM containing 10% fetal bovine serum (FBS) (Sigma-Aldrich, F4135), 1% penicillin/streptomycin (Gibco, 15070-063), and 25 µM L-glutamine (Gibco, 25030081). Following 2 hr in culture, media was replaced with supplemented FBS. At 24 hr after plating, media was replaced with Neurobasal-A (NBA) (Life Technologies, 10888-022), 2% B27 (Gibco, 17504-044), and 10 µM uridine/fluoro-deoxyuridine (Sigma-Aldrich, U3003/F0503). Neurons were maintained in a 37°C humidified incubator with 5% $CO_2$.

## Neuron immunocytochemistry

DIV1 or DIV2 primary cortical neurons were washed with phosphate-buffered saline (PBS) (Gibco, 10010023) followed by PHEM buffer, which contains 60 mM PIPES (Sigma-Aldrich, P6757), 25 mM HEPES (Sigma-Aldrich, H3375), 10 mM EGTA (Sigma-Aldrich, E3889), and 2 mM $MgCl_2$ (Acros, 223211000). Soluble tubulin dimers were extracted using 0.1% Triton-X (Sigma-Aldrich, T9284) with 10 µM taxol (LC Labs, P-9600) and 0.1% DMSO (Fisher, M-1739) in PHEM buffer. Cells were fixed with 4% PFA (Sigma-Aldrich, 158127) and 0.1% glutaraldehyde (Sigma-Aldrich, G7776) in PBS for 10 min at room temperature, washed with PBS, then blocked with blocking buffer, 3% BSA (Fisher, 50-253-893) and 0.2% Triton-X in PBS. Cells were reduced with 10 mg/ml sodium borohydride (Fisher, S678-10) in an equal mixture of PBS and methanol (Acros, 177150050) for 7 min at room temperature, then washed three times with PBS. Cells were incubated with blocking buffer for 20 min at room temperature. Immunostaining was performed using a primary antibody directed against 6X-His (Invitrogen, 4A12E4 37-2900; 1:500), β-II tubulin (Abcam, ab179512; 1:500), acetylated tubulin (Sigma, T7451; 1:1000), tyrosinated tubulin (Sigma, MAB1864-I; 1:1000). Primary antibody was diluted in blocking buffer and incubated overnight at 4°C in a humidified chamber. After primary antibody staining, cells were washed three times with PBS. The secondary antibodies goat anti-rabbit IgG Alexa Fluor 568 (Thermo, A-11011), goat anti-rat IgG Alexa Fluor 568 (Thermo, A-11077), or goat anti-mouse IgG Alexa Fluor 647 (Thermo, A32728) were diluted 1:200 in blocking buffer and 1% normal goat serum (Vector Laboratories, S-1000-20) and incubated for 2 hr at room temperature in a dark container. Cells were sealed with glass coverslips and aqueous mounting media containing DAPI (Vector Laboratories, H-1200), then imaged on a spinning disk confocal microscope with a 40× oil objective (see details below). Statistical analysis between multiple groups were analyzed by two-way ANOVA and analyzed post hoc by Tukey test.

## Neuron cold temperature shock

The media of DIV1 neuron cultures (as described above) was replaced with Neurobasal-A (NBA) (Life Technologies, 10888-022), 2% B27 (Gibco, 17504–044), and 10 µM uridine/fluoro-deoxyuridine (Sigma-Aldrich, U3003/F0503). The dishes were then placed at 4°C for the desired time. At each time point, dishes were removed from 4° C and placed on ice while following the fixing protocol described above.

## Neuron morphology and PTMs analysis

All images were analyzed using ImageJ (National Institute of Health) and the NeuronJ plugin (*Meijering et al., 2004*). Neuron processes were defined as primary, secondary, or tertiary, then quantified for

each cell analyzed. PTM fluorescent signals were quantified from each cell by measuring line scans along processes. For axons, the data were binned by 0–10 and 10–20 µm from the distal axonal tip. Similarly for dendrites, the data were by binned by 0–5 and 5–10 µm from the distal dendritic tip. The dendrite used for analysis was chosen as the one immediately counter-clockwise of the axon. For consistency within each individual cell, the line scans were first traced using cytoplasmic GFP signal, saved, and then used to measure signal from the immunofluorescent staining for the tubulin PTMs in different channels. A two-way ANOVA analyzed post hoc by Tukey test was used for statistical analyses between multiple groups.

## Neuron microtubule dynamics

Cortical neuron cultures were transfected with pCIG2 plasmids co-expressing *TUBA1A* and GFP-MACF43 as described above. DIV2 cells were kept at 37°C and 5% $CO_2$ via a LiveCell incubator (Pathology Devices, Inc) during image acquisition. Images were acquired on an inverted microscope (Ti Eclipse) with a 100× Plan-Apochromat (NA 1.43) objective lens (Nikon) and a spinning disk (Yokogawa X1). Images were captured with a charge-coupled device camera (iXon X3; Andor Technology). All images were acquired with SlideBook (3i). A Z-series covering a 5 µm range at 0.5 µm steps was acquired every 2.2 s for 4 min. MACF43-comets were analyzed using the UTrack software (*Applegate et al., 2011*). Statistical analysis was done using an unpaired t-test.

## Yeast microtubule dynamics

Microtubule dynamics were analyzed in log-phase, pre-anaphase yeast cells in which all the α-tubulin in the cell was either WT, V410I, or V410A. The mutants were integrated at the native *TUB1* and *TUB3* loci. Microtubule plus ends were identified by Bik1-3GFP. Images were collected on a Nikon Ti-E microscope equipped with a 1.45 NA 100× CFI Plan Apo objective, piezo electric stage (Physik Instrumente, Auburn, MA), spinning disk confocal scanner unit (CSU10; Yokogawa), 488 and 561 nm lasers (Agilent Technologies, Santa Clara, CA), and an EMCCD camera (iXon Ultra 897; Andor Technology, Belfast, UK) using NIS Elements software (Nikon). Cells were grown asynchronously to early log-phase in non-fluorescent media and adhered to slide chambers coated with concanavalin A. Slide chambers were sealed with VALAP (Vaseline, lanolin, and paraffin at 1:1:1). Z-series consisting of a 7 µm range at 0.4 µm steps were acquired every 4 s for 10 min at 30°C. Astral microtubule lengths were measured in each frame as the distance from the spindle pole body to the microtubule plus end. Cell genotypes were blinded for analysis. Statistical analysis was done using a one-way ANOVA followed by a Tukey test to correct for multiple comparison tests.

## Stu2 localization

Analysis was done in log-phase, pre-anaphase budding yeast cells expressing either *tub1/tub3*-WT, V410I, or V410A, as well as Stu2-3GFP. The tubulin mutations and the fluorescent tagging of Stu2 were done at the endogenous loci. Short time-lapse images, every 5 s for 30 s, were collected on a spinning disc confocal microscope with a 100× oil objective, using Z-series consisting of 0.4 µm steps over a total range of 7 µm. Stu2-3GFP fluorescence intensity was measured at the plus ends of growing astral microtubules by defining a 182 $nm^2$ region. A region of the same size and adjacent to the Stu2-3GFP foci was also measured and subtracted from the Stu2-3GFP intensity value to account for background fluorescence. Statistics represent values from a one-way ANOVA corrected for multiple comparison by a Tukey test.

## GST-TOG1/2 purification

Purification of GST-TOG1/2 followed previously described methods and is described briefly below (*Reusch et al., 2020*; *Widlund et al., 2012*). The pGEX-6P-1 Stu2 1–590 plasmid was transformed into BL21 cells and colonies were grown on an LB plate containing 100 µg/ml carbenicillin and 15 µg/ml chloramphenicol at 37°C. A colony was inoculated overnight at 37°C in MDAG-135 media (25 mM $Na_2HPO_4$, 25 mM $KH_2PO_4$, 50 mM $NH_4Cl$, 5 mM $Na_2SO_4$, 2 mM $MgSO_4$, 0.2× metals, 0.35% glucose, 0.1% aspartate, 200 µg/ml each of 18aa (no C, Y)) containing 100 µg/ml carbenicillin and 15 µg/ml chloramphenicol (*Studier, 2005*). The following day the colony was diluted 500-fold in 1 l of Terrific Broth media (Fisher, BP2468-2) with 100 µg/ml carbenicillin and 15 µg/ml chloramphenicol and grown at 37°C until the $OD_{600}$ reached approximately 0.5. The cultures were shaken at 18°C for 1 hr, then

induced with 0.2 mM isopropyl β-$_D$-1-thiogalactopyranoside for 18 hr (Fisher, BP1620-1). Following induction cells were pelleted at 6200× $g$ for 10 min at 6°C and resuspended in a 1:1 ratio with a buffer consisting of 2× PBS, 1 mM dithiothreitol (DTT), 20 µl benzonase (25 U/µl), and 1× protease inhibitors (2 cOmplete EDTA-free Tablets per 100 ml; Roche, 04693132001). The cells were lysed by two passes through a microfluidizer at 1500 bar. Cell lysate was then clarified by spinning at 12,000 rpm for 30 min at 4°C.

A 5 ml GSTrap HP column (GE, 17-5282-01) was pre-equilibrated with 5 column volumes (CV) of wash buffer (2× PBS, 1 mM DTT) at 1 ml/min. The clarified lysate was then loaded onto the column at 0.5 ml/min. The following washes were all done at 1 ml/min. The column was washed with 10 CV of wash buffer with 0.1% Tween 20, then 2 CV of 2× PBS with 5 mM ATP and 10 mM MgCl$_2$. The column incubated in this buffer for 20 min. The column was then washed with 5 CV of 6× PBS, followed by 5 CV of wash buffer. GST-TOG1/2 was eluted from the column in 1 ml fractions into a deep 96-well plate with wash buffer plus 5 mM reduced glutathione, pH 8.0. The eluted fractions were run on a gel and stained with Coomassie. Fractions containing GST-TOG1/2 were pooled and dialyzed against three changes (after 2 hr, overnight, then another 2 hr) of coupling buffer (100 mM NaHCO$_3$, 100 mM NaCl, pH 8.2) at 4°C. The dialyzed sample was concentrated in 0.5 ml centrifuge concentrating filters (Sigma Z677108) that had been pre-equilibrated with coupling buffer. GST-TOG1/2 was concentrated to 14.5 µM and stored in 25 µl aliquots at –80°C.

## Yeast tubulin purification

Purification of yeast tubulin was based on previously described methods with slight modifications (*Johnson et al., 2011*). Protease deficient yeast cells (JEL1) were transformed with pRS426-GAL-Tub1-internal 6X-His and pRS424-GAL-Tub2 (untagged) (*Lindsley and Wang, 1993*). tub1-V410I and -V410A mutants were constructed in the pRS426-GAL-Tub1-internal 6X-His plasmid with QuikChange XL. Cells were grown in five or six 5 ml cultures of selection media (-ura -trp) supplemented with 2% glucose for either 3 or 2 days, respectively, at 30°C. Five ml cultures were then transferred to 50 ml cultures of the same media to grow for 24 hr. Cultures were then transferred to 1 l cultures of YPGL, consisting of 10 g yeast extract (Fisher, BP1422), 20 g peptone (Fisher, BP1420), 30 ml glycerol (Fisher, G33), and 33 ml lactate (Sigma, L1375). Yeast extract, peptone, and 850 ml ddH$_2$O was combined and autoclaved the day before it was needed. The glycerol and lactate were added immediately after media came out of the autoclave, then allowed to cool at room temperature overnight. After 20–24 hr of growing in 1 l cultures and the OD$_{600}$ was between 5.0 and 9.0, 1 l of cells was induced with 20 g galactose (Chem-Impex, 01449) for 5 hr. Cells were spun down at 4°C for 15 min at 4000 rpm (3040 RCF) in a J6 centrifuge (Beckman Coulter Life Sciences). Supernatant was discarded and pellets were stored in 15 ml conical tubes in –80°C. This process was done three times for a total of approximately 100 g of cells before moving on to purification.

Cell pellets were thawed on ice and combined with cold lysis buffer (50 mM HEPES, 500 mM NaCl, 10 mM MgSO$_4$, 30 mM imidazole) plus freshly added 50 µm GTP (Sigma G8877) and 1 mM PMSF (Acros Organics, 215740050) to a total of approximately 150 ml. While cells were thawing, the microfluidizer (Microfluidics, M-110P) was packed with ice to cool. Cells were passed through the microfluidizer four times at 27,000 PSI with 10 min on ice between each pass. The cells were chased with 50 ml lysis buffer supplemented with 50 µM GTP and 1 mM PMSF. The lysed cells were clarified in an Avanti J-26S XPI centrifuge (Beckman Coulter Life Sciences) at 41,000 RCF for 30 min at 4°C. The pellets were discarded and the supernatant was applied to a 5 ml His60 Ni column (Takara, 635680) that was pre-equilibrated with 20 CV of lysis buffer. The column was washed with 10 CV of lysis buffer supplemented with 50 µM GTP, then with 10 CV of wash buffer (25 mM PIPES pH 6.9, 1 mM MgSO$_4$, 30 mM imidazole, 200 mM NaCl) with 50 µM GTP. Protein was eluted from the column with 10 CV of elution buffer (25 mM PIPES pH 6.9, 1 mM MgSO$_4$, 300 mM imidazole, 200 mM NaCl) with 50 µM GTP into 2 × 5 ml samples followed by 4 × 10 ml samples. Samples from the wash and elution steps were run on a gel and Coomassie staining was used to determine which fractions contained tubulin. Elution fractions with tubulin were pooled and 20% glycerol was added before flash freezing the samples in liquid nitrogen and storing at –80°C.

Elution fractions from the Ni column were thawed in a room temperature water bath and 10 µl of nuclease (Pierce, 88701) was added per 25 ml of sample, then sat for 1 hr at room temperature. Two HiTrap Q HP 1 ml columns (GE, 17-1153-01) were strung together on an AKTA FPLC (GE) and

equilibrated with 10 CV (20 ml) of ddH$_2$O, 5 CV of Buffer A (25 mM PIPES pH 6.9, 2 mM MgSO$_4$, 1 mM EGTA), 5 CV of Buffer B (25 mM PIPES pH 6.9, 2 mM MgSO$_4$, 2 mM EGTA, 1 M NaCl), and 10 CV of Buffer A, all run at 1 ml/min. The column was then equilibrated at 2 ml/min with 12.5 CV of 80% Buffer A and 20% Buffer B, both supplemented with 50 µM GTP. Sample from Ni column was injected into a 50 ml Superloop that was washed with diH$_2$O followed by Buffer A. The sample was applied to the column at 0.5 ml/min using Buffer A. Once the sample was loaded the tubulin was eluted with 30 CV of Buffer A and a 20–60% gradient of Buffer B, both supplemented with 50 µM GTP. Elution was collected in 0.5 ml fractions in a 96 deep-well plate. Elution fractions were run on a gel and Coomassie staining determined which fractions had pure tubulin and would be used to concentrate the protein.

Amicon ultra 0.5 ml centrifuge concentrating filters with a 10 kDa cutoff (Sigma, Z677108) were prepared with two rounds of 0.4 ml 1% Triton X-100, ddH$_2$O, and PEM (0.1 M PIPES, 1 mM EGTA, 1 mM MgCl$_2$) spinning at 800 RCF for 5 min at 4°C. Up to 0.450 ml of pure tubulin sample was applied to the concentrating filters and spun at 4°C at 800 RCF for anywhere between 2 and 5 min, depending on the concentration of the sample. The concentration was checked after each spin on a Take3 Micro-volume Plate (BioTek) using 260 and 280 nm absorbance to ensure that the concentration was increasing but aggregates were not forming. Samples were constantly kept on ice or at 4°C and concentrated to 2.2 µM (tub1-V410I), 2.3 µM (tub1-V410A), and 3 µM (TUB1). Concentrated protein samples were run through a Zeba spin 2 ml desalting column (Thermo, 89890) that was prepared by spinning at 1000 RCF for 2 min at 4°C to remove storage buffer, then washed three times with 0.4 ml PEM. Up to 0.7 ml of concentrated sample was added to each desalting column, then spun down at 1000 RCF for 2 min at 4°C. The flow-through sample was supplemented with 50 µM GTP, aliquoted into 20 µl samples that were flash frozen in liquid nitrogen, and stored at –80°C. At least two separate rounds of purifications were used as technical replicates for in vitro assays that used this purified tubulin.

## TOG binding assay

Pierce Glutathione Magnetic Agarose Beads (Thermo, 78602) were used to separate purified GST-TOG1/2 (above) and anything bound in complex with TOG1/2 from the rest of the reaction; 100 µl of bead slurry (equates to 25 µl settled beads) were equilibrated twice with 40 µl wash buffer (125 mM Tris-HCl, 150 mM KCl, 1 mM DTT, 1 mM EDTA, pH 7.4) by vortexing for 10 s and removing buffer by placing tube on magnetic stand. In separate tubes, 100 nM tubulin (WT, V410I, or V410A; purification described above) was mixed with a range of GST-TOG1/2 concentrations (from 9.8 nM to 5 µM) at 4°C for 15 min. Reaction was added to the equilibrated beads and rotated at 4°C for 1 hr. Tubes were placed on magnetic stand and supernatant was collected. The supernatant was prepared with Laemmli sample buffer and β-mercaptoethanol and loaded onto a 10% SDS-PAGE gel. The gel was transferred to a PVDF membrane (Millipore, IPFL85R), blocked with a PBS Blocking Buffer (LI-COR, 927–70001) at room temperature for 1 hr, and probed with mouse anti-α-tubulin (4A1; 1:100) or mouse anti-β-tubulin (E7; 1:100) overnight at 4°C. The following day the membranes were washed once for 5 min with 1× PBS, then incubated at room temperature in the dark for 1 hr with goat anti-mouse-680 (LI-COR, 926-68070; 1:15,000). Membranes were washed twice with 1× PBST, then once with 1× PBS before imaging on an Odyssey Imager (LI-COR, 2471). Both α- and β-tubulin band intensity was analyzed using Image Studio v5.2 (LI-COR). Tubulin band intensity represents the amount of tubulin in solution that is not bound to GST-TOG1/2.

## Stu2 depletion and yeast immunofluorescence

Stu2 depletion strains originally constructed from Kosco et al. were graciously gifted from Dr Tim Huffaker (*Kosco et al., 2001*). Strains were modified to mutate the endogenous *TUB1* locus to either V410I or V410A using site-directed mutagenesis. Cells were grown to log-phase, then induced with 500 µM cupric sulfate for 1.5 hr at 30°C. The following fix and staining protocol was adapted from *Miller, 2004*, as described. Cells were fixed with 3.7% formaldehyde (Sigma-Aldrich, 252549) at 30°C for 2 hr, then spun down at 3000 rpm for 3 min. Pellet was washed twice with wash buffer (40 mM KPO$_4$, pH 6.5), then stored overnight at 4°C. The pellet was washed twice with wash buffer plus 1.2 M sorbitol. Fixed cells were digested with 10 µl of 20T 50 mg/ml zymolyase (Nacali Tesque, 07663-91) and 15 µl β-mercaptoethanol (Sigma-Aldrich, M3148) for 45 min at 37°C. Digested cells were spun down at 2000 rpm for 3 min, washed once with wash buffer plus 1.2 M sorbitol, then resuspended in

20 µl wash buffer plus 1.2 M sorbitol. Twenty µl of cells were spotted onto each well of a Teflon-coated 10-well slide (Polysciences, 18357) that had been pre-treated with 10 ng/µl poly-L-lysine. Cells adhered for 10 min at room temperature. Liquid was aspirated off before immediately permeabilizing the cells in a coplin jar of cold methanol for 6 min, followed by immersing the slide in cold acetone for 30 s. Cells were blocked at room temperature for 1 hr in blocking buffer (1× PBS + 0.5% BSA), then incubated overnight at 4°C in a humid chamber with mouse anti-α-tubulin (4A1; 1:100 in blocking buffer). Wells were washed four times for 10 min with blocking buffer, then incubated at room temperature for 1 hr in a dark humid chamber with goat anti-mouse IgG-Alexa488 (Invitrogen, A11001). Cells were washed four more times with blocking buffer. DAPI mounting solution (Vector Laboratories, H-1200) was added to the slide, then imaged on widefield microscope with a 100× oil objective. Statistical analysis was done using a two-way ANOVA corrected for multiple comparison by a Tukey test.

### Yeast growth assays

For liquid growth assays, saturated cultures were diluted 1:500 in rich media. In a 96-well plate 200 µl/ well was used. $OD_{600}$ values were measured on an Epoch 2 Microplate Spectrophotometer (BioTek #EPOCH2NS) every 5 min for at least 19 hr at 30°C with orbital shaking. The doubling time was calculated by fitting the measured $OD_{600}$ values to an exponential curve using a MATLAB code that has previously been described (*Fees et al., 2016*). The doubling times have been normalized to WT cells. A two-way ANOVA was used to compare across multiple groups and corrected post hoc by a Tukey test.

### 4°C temperature shock in yeast

Cells expressing either TUB1/TUB3, tub1/tub3-V410I, or tub1/tub3-V410A along with GFP-Tub1, GFP-Tub1-V410I, or GFP-Tub1-V410A, respectively, were grown overnight at 30°C in rich liquid media. The following day the cells were diluted in fresh media and grown to log-phase. The log-phase cultures were then transferred to a shaker at 4°C for the indicated amount of time. Cells were fixed at 4°C with 3.7% formaldehyde and 0.1 M $KPO_4$ for 3 min. Cells were pelleted on a desktop centrifuge and supernatant was discarded. The pellet was resuspended in quencher solution (0.1% Triton-X, 0.1 M $KPO_4$, 10 mM ethanolamine). The cells were pelleted and washed twice with 0.1 M $KPO_4$. Imaging chamber slides were coated with concanavalin A and washed with 0.1 M $KPO_4$ before loading fixed cells into the chambers and sealed with VALAP (Vaseline, lanolin, and paraffin at a 1:1:1 ratio) (*Fees and Moore, 2018*).

### In vitro dynamics analysis using IRM

IRM was used to measure microtubule dynamics of unlabeled purified tubulin. Chambers were assembled with two different sized coverslips, 22 mm × 22 mm and 18 mm × 18 mm, that were silanized and plasma cleaned. Three strips of single ply parafilm were melted between the coverslips to create two chambers on a custom-made stage insert. Anti-rhodamine antibody (Life Technologies, A6397) was diluted 1:100 in cold BRB80 (80 mM PIPES, 1 mM $MgSO_4$, 1 mM EGTA), flowed into the chamber, then sat at room temperature for 5 min. The antibody was then washed out with room temperature BRB80. Chambers were flushed with 1% pluornic-F127 in BRB80 and incubated in this buffer for 5 min, then washed out with room temperature BRB80. The stage insert was placed in the scope enclave to equilibrate at 30°C for 30 min. GMPCPP-stabilized microtubule seeds that attached to the anti-rhodamine antibody on the coverslips were flowed into the chamber where they settled for 30 s before washing out with room temperature BRB80. The chambers were washed with reaction buffer that contained no tubulin (1× PEM, 0.1% methyl cellulose, 1 mM GTP, 0.1 mg/ml BSA) before adding the reaction buffer along with 0.3–0.9 µM tubulin. The chamber was then sealed with VALAP. The reaction in the chamber was equilibrated on the widefield microscope for 20 min at 30°C to prevent drifting during imaging. Following the equilibration period, images were collected every second for 1 hr in the IRM channel (*Mahamdeh et al., 2018*). The 561 nm channel was also collected for the first frame to mark the GMPCPP-stabilized seeds. ImageJ was used for analyzing microtubule dynamics by generating kymographs of individual microtubules.

## Acknowledgements

We are grateful to members of the Moore and Dr Emily Bates (CU AMC) labs for helpful discussions. We thank the lab of Dr Matthew Kennedy (CU AMC) for providing P0-P2 rat cortex, and Adam Soh, Alex Stemm-Wolf, and Dr Chad Pearson (CU AMC) for assistance with imaging microtubule dynamics in living neurons. This work was supported by T32 GM136444 (KJH); Center for Neuroscience Pilot Award from the Colorado Clinical and Translational Sciences Institute (funded by NIH UL1TR002535), Boettcher Webb-Waring Biomedical Research Program Award from the Boettcher Foundation, and the Children's Hospital Colorado Program in Pediatric Stem Cell Biology (SJF); and NIH R35GM136253 (JKM).

## Additional information

### Funding

| Funder | Grant reference number | Author |
|---|---|---|
| National Institutes of Health | Graduate Student Fellowship, T32GM136444 | Katelyn J Hoff |
| National Institutes of Health | R35GM136253 | Jeffrey K Moore |
| Colorado Clinical and Translational Sciences Institute | Center for Neuroscience Pilot Award | Santos J Franco |
| Boettcher Foundation | Boettcher Webb-Waring Biomedical Research Program Award | Santos J Franco |
| Children's Hospital Colorado | Program in Pediatric Stem Cell Biology | Santos J Franco |

The funders had no role in study design, data collection and interpretation, or the decision to submit the work for publication.

### Author contributions

Katelyn J Hoff, Conceptualization, Data curation, Formal analysis, Funding acquisition, Investigation, Methodology, Validation, Visualization, Writing - original draft, Writing - review and editing; Jayne E Aiken, Conceptualization, Formal analysis, Investigation, Methodology, Writing - review and editing; Mark A Gutierrez, Data curation, Investigation, Methodology; Santos J Franco, Methodology, Project administration, Resources, Writing - review and editing; Jeffrey K Moore, Conceptualization, Funding acquisition, Investigation, Methodology, Project administration, Resources, Supervision, Writing - original draft, Writing - review and editing

### Author ORCIDs

Katelyn J Hoff http://orcid.org/0000-0002-2678-1859
Jeffrey K Moore http://orcid.org/0000-0003-3262-3248

### Ethics

Animals were housed and maintained according to protocols (#00019) approved by the Institutional Animal Care and Use Committee of the University of Colorado Anschutz Medical Campus (PHS Animal Assurance of Compliance #D16-00171). All surgeries were performed under inhaled vaporized isoflurane anesthesia with pre- and post-operative analgesic (meloxicam). Every effort was made to minimize pain and suffering.

### Decision letter and Author response

Decision letter https://doi.org/10.7554/eLife.76189.sa1
Author response https://doi.org/10.7554/eLife.76189.sa2

## Additional files

### Supplementary files
• Transparent reporting form

### Data availability
All data generated or analysed during this study are included in the manuscript and supporting file; Source Data files have been provided.

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

# Appendix 1

**Appendix 1—table 1.** Plasmids.

| Plasmid ID | Plasmid name | Source |
|---|---|---|
| pJM0157 | pGEX6p1-GST-STU2 1–590 | *Widlund et al., 2012* |
| pJM0295 | pCIG2 | *Hand et al., 2005* |
| pJM0288 | pRS426-GAL-Tub1-internal 6X-His | *Johnson et al., 2011* |
| pJM0289 | pRS424-GAL-Tub2 (untagged) | *Johnson et al., 2011* |
| pJM0430 | pCIG2-Tuba1a(WT)-IRES-GFP-MACF43 | *Aiken et al., 2019* |
| pJM0459 | Stu2-3GFP | This study |
| pJM0469 | TUB1-pRS306 integrating plasmid | *Aiken et al., 2020* |
| pJM0477 | pCIG2-Tuba1a(V409A)-IRES-GFP-MACF43 | This study |
| pJM0507 | Tub1-V410I-pRS306 integrating plasmid | This study |
| pJM0510 | pCIG2-Tuba1a(V409I)-IRES-GFP-MACF43 | This study |
| pJM0560 | Tub1-V410A-pRS306 integrating plasmid | This study |
| pJM0546 | pCIG2-TUBA1A-6X-His | *Buscaglia et al., 2020a* |
| pJM0558 | pRS426-GAL-Tub1(V410A)-internal 6X-His | This study |
| pJM0559 | pRS426-GAL-Tub1(V410I)-internal 6X-His | This study |
| pJM0612 | pCIG2-TUBA1A(V409I)-int-6X-His | This study |
| pJM0642 | pCIG2-TUBA1A(V409A)-int-6X-His | This study |

**Appendix 1—table 2.** Yeast strains.

| Yeast strain | Genotype | Source |
|---|---|---|
| yJM3733 | *MATα tub1::pJM469(TUB1 + 487::URA) STU2-3GFP::TRP ura3-52 lys2-801 leu2-Δ1 his3-Δ200 trp1-Δ63* | This study |
| yJM3734 | *MATa tub1::pJM469(TUB1 + 487::URA) STU2-3GFP::TRP ura3-52 lys2-801 leu2-Δ1 his3-Δ200 trp1-Δ63* | This study |
| yJM4010 | *MATa BIK1-3GFP::TRP1 tub1::pJM560(TUB1-V410A+487::URA) tub3-V410A+150::hygB ura3-52 lys2-801 leu2-Δ1 his3-Δ200 trp1-Δ63* | This study |
| yJM4011 | *MATα BIK1-3GFP::TRP1 tub1::pJM560(TUB1-V410A+487::URA) tub3-V410A+150::hygB ura3-52 lys2-801 leu2-Δ1 his3-Δ200 trp1-Δ63* | This study |
| yJM4014 | *MATa BIK1-3GFP::TRP1 tub1::pJM507(TUB1-V410I+487::URA) tub3-V410I+150::hygB ura3-52 lys2-801 leu2-Δ1 his3-Δ200 trp1-Δ63* | This study |
| yJM4015 | *MATα BIK1-3GFP::TRP1 tub1::pJM507(TUB1-V410I+487::URA) tub3-V410I+150::hygB ura3-52 lys2-801 leu2-Δ1 his3-Δ200 trp1-Δ63* | This study |
| yJM4077 | *MATa STU2-3GFP::TRP1 tub1::pJM560(TUB1-V410A+487::URA) tub3-V410A+150::hygB ura3-52 lys2-801 leu2-Δ1 his3-Δ200 trp1-Δ63* | This study |
| yJM4078 | *MATα STU2-3GFP::TRP1 tub1::pJM560(TUB1-V410A+487::URA) tub3-V410A+150::hygB ura3-52 lys2-801 leu2-Δ1 his3-Δ200 trp1-Δ63* | This study |
| yJM4081 | *MATa STU2-3GFP::TRP1 tub1::pJM507(TUB1-V410I+487::URA) tub3-V410I+150::hygB ura3-52 lys2-801 leu2-Δ1 his3-Δ200 trp1-Δ63* | This study |
| yJM4082 | *MATα STU2-3GFP::TRP1 tub1::pJM507(TUB1-V410I+487::URA) tub3-V410I+150::hygB ura3-52 lys2-801 leu2-Δ1 his3-Δ200 trp1-Δ63* | This study |
| yJM4726 | *MATα tub1::pJM469(TUB1+487::URA) TUB3+150::hygB BIK1-3GFP::TRP1 ura3-52 lys2-801 leu2-Δ1 his3-Δ200 trp1-Δ63* | This study |
| yJM4727 | *MATa tub1::pJM469(TUB1+487::URA) TUB3+150::hygB BIK1-3GFP::TRP1 ura3-52 lys2-801 leu2-Δ1 his3-Δ200 trp1-Δ63* | This study |

*Appendix 1—table 2 Continued on next page*

*Appendix 1—table 2 Continued*

| Yeast strain | Genotype | Source |
|---|---|---|
| yJM4835 | *MATa PACE1-UBR1 PACE1-ROX1 stu2Δ::URA3::PANB1-UB-R-STU2 ura3-52 trp1-Δ1 ade2-101 lys2-801* | **Kosco et al., 2001** |
| yJM4836 | *MATa PACE1-UBR1 PACE1-ROX1 ura3-52::URA3 ura3-52 trp1-Δ1 ade2-101 lys2-801* | **Kosco et al., 2001** |
| yJM4859 | *MATa PACE1-UBR1 PACE1-ROX1 stu2Δ::URA3::PANB1-UB-R-STU2 tub1::pJM754(TUB1+487::TRP1) ura3-52 trp1-Δ1 ade2-101 lys2-801* | This study |
| yJM4860 | *MATa PACE1-UBR1 PACE1-ROX1 stu2Δ::URA3::PANB1-UB-R-STU2 tub1-V410I::pJM755(TUB1+487-V410I::TRP1) ura3-52 trp1-Δ1 ade2-101 lys2-801* | This study |
| yJM4861 | *MATa PACE1-UBR1 PACE1-ROX1 stu2Δ::URA3::PANB1-UB-R-STU2 tub1-V410A::pJM756(TUB1+487-V410A::TRP1) ura3-52 trp1-Δ1 ade2-101 lys2-801* | This study |

## Appendix 1—table 3. Oligo sequences (5′ to 3′).

| # | Name | Application | Sequence | Species | Direction |
|---|---|---|---|---|---|
| 674 | TUBA1A-V409A F QC | TUBA1A-V409A F Quikchange | ctttgttcactggtacgctggggaggggatg | Human | Forward |
| 675 | TUBA1A-V409A R QC | TUBA1A-V409A R Quikchange | catcccctccccagcgtaccagtgaacaaag | Human | Reverse |
| 869 | MACF43 + NotI R | PCR up MACF43 and clone into pCIG2 | tatc GCGGCCGC TCATCTCTTTGAGGACTTGTCC | Mammalian | Reverse |
| 870 | BsrGI + MACF43 F | PCR up MACF43 and clone into pCIG2 | GAGC TGTACA AGTCCGGCCGGACTCAG | Mammalian | Forward |
| 986 | TUBA1A-V409I F QC | TUBA1A-V409I F Quikchange | cctttgttcactggtacattggggaggggatgg | Human | Forward |
| 987 | TUBA1A-V409I R QC | TUBA1A-V409I R Quikchange | ccatcccctccccaatgtaccagtgaacaaagg | Human | Reverse |
| 1042 | TUBA1A Gibson PstI insert F | Amplify TUBA1A from pINCY into pCIG2 | ctcaagcttcgaattctgcaATGcgtgagtgcatctccatcc | Mouse | Forward |
| 1043 | TUBA1A Gibson PstI insert R | Amplify TUBA1A from pINCY into pCIG2 | cggtaccgtcgactgcaTTAgtattcctctccttcttcctcaccc | Mouse | Reverse |
| 1075 | TUBA1A-T349E F QC | TUBA1A-T349E F Quikchange | gtttgtggattggtgccccgagggcttcaaggttggcatca | Human | Forward |
| 1076 | TUBA1A-T349E R QC | TUBA1A-T349E R Quikchange | tgatgccaaccttgaagccctcggggcaccaatccacaaac | Human | Reverse |
| 867 | pCIG2-TUBA1A stop +53 R | TUBA1A sequencing | GGCTTCGGCCAGTAACGTTAG | Mammalian | Reverse |
| 868 | pCIG2-TUBA1A start – 52 F | TUBA1A sequencing | CCTACAGCTCCTGGGCAACG | Mammalian | Forward |
| 545 | tub1-V410A F | tub1-V410A F Quikchange | CGTCCACTGGTATGCCGGTGAAGGTATG | Yeast | Forward |
| 546 | tub1-V410A R | tub1-V410A R Quikchange | CATACCTTCACCGGCATACCAGTGGACG | Yeast | Reverse |
| 1018 | tub1-V410I F | tub1-V410I F Quikchange | cgtgctttcgtccactggtatatcggtgaaggt | Yeast | Forward |
| 1019 | tub1-V410I R | tub1-V410I R Quikchange | accttcaccgatataccagtggacgaaagcacg | Yeast | Reverse |
| 1280 | tub3-V410A F | tub3-V410A F to introduce mutation at TUB3 locus | TGATGTATGC CAAACGTGCT TTCGTCCATT GGTAT GcC GGTGAAGGTATG GAAGAAGGTG | Yeast | Forward |
| 1281 | tub3-V410I F | tub3-V410I F to introduce mutation at TUB3 locus | TGATGTATGC CAAACGTGCT TTCGTCCATT GGTAT aTC GG TGAAGGTATG GAAGAAGGTG | Yeast | Forward |
| 5551 | TUB3 + 154 R | To introduce V410 mutations at TUB3 locus | GCTATTGTTCTAACTGTACACCTAAAGGTAAGTG gca tag gcc act agt gga tc | Yeast | Reverse |

