## [Editor Report]

Tubulin mutations underlie a number of neurodevelopmental diseases, but their molecular effects remain largely unknown. Using a combination of approaches and model systems, Hoff et al., provide evidence that disease-associated α-tubulin mutations increase microtubule polymerization rates and block the recruitment of a regulatory binding protein, which combinatorially promotes excessive neurite branching that disrupts neuronal migration. Overall, this study demonstrates a link between the regulation of microtubule dynamics and disease pathogenesis.

---

## [Decision Letter]

**Decision letter after peer review:**

Thank you for submitting your article "Tubulinopathy mutations in *TUBA1A* that disrupt neuronal morphogenesis and migration override XMAP215/Stu2 regulation of microtubule dynamics" for consideration by *eLife*. Your article has been reviewed by 3 peer reviewers, and the evaluation has been overseen by a Reviewing Editor and Anna Akhmanova as the Senior Editor. The reviewers have opted to remain anonymous.

There is general agreement from the reviewers that the yeast data is rigorous, soundly interpreted, and represents the core of the manuscript. However, there are concerns about the interpretation of the neuronal data and the conclusion that the mutations affect dimer conformation. It is therefore suggested that the authors expand the neuronal data, since performing structural studies on tubulin dimer conformation is outside the scope of this paper. In addition, there are many areas where it is suggested that the authors soften their wording when there is no direct evidence to support their conclusions. In particular, the reviewers strongly recommend the authors remove/extensively limit all the discussion of the enhancement of tubulin straightening for which there is no convincing data. Please read the individual reviewer comments for more information.

Essential revisions:

1) To determine if the results from their yeast model that the V409 mutations have a strong effect on microtubule polymerization are consistent in neurons, the authors should assay directly for an effect on microtubule polymerization in neurons using a +TIP marker, such as EB3-GFP.

2) As suggested by Reviewer 1, please define the neurites you are analyzing. Even if you cannot assign them as axons or dendrites, please define them as primary, secondary, tertiary, etc., as is common across the neurobiology field.

3) Both Reviewer 1 and 3 expressed concerns about the analysis and conclusions of the neurite retraction assays. Please address both comments, but especially please see Reviewer 1's helpful third comment about convincing ways to analyze neurite retraction.

4) Both Reviewer 1 and 3 noted that conclusions about morphology transitions are not supported by the data presented in the paper. Please either address this experimentally as suggested by the reviewers, or remove this conclusion. The authors should also address the other possibilities such as a delay in neuronal migration, as suggested by Reviewer 3.

5) During the discussion, all three reviewers raised concerns about the lack of data and over-interpretation for how the mutants relate to the conformation of the tubulin dimer. At the very least, the authors should provide a structure of the tubulin dimer with the residues highlighted. This can easily be done in pymol or a similar program. Perhaps once the authors include this data, they may modify their conclusions on whether these mutations could feasibly affect dimer conformation, or if, based on the residue locations, there are other models that the authors may want to discuss. Reviewer 2 has the most thorough review on this point.

6) It was agreed in the reviewer discussion that the authors should discuss potential effects on the total amount of tubulin available for polymerization if TOG-tubulin binding is inactivated. This could be the reason for the enhancement in microtubule polymerization rates observed with the mutants and would be consistent with most of the data.

7) Finally, please provide precise n for the number of cells per condition and the number of replicates. In addition, please present the statistical tests in the legends, as is outlined in the *eLife* guide to authors.

*Reviewer #1 (Recommendations for the authors):*

1. Given the findings from their yeast model that the V409 mutations have a strong effect on microtubule polymerization, the authors should assay directly for an effect on microtubule polymerization in neurons (e.g. using EB3-GFP).

2. The authors find that yeast V410 mutants are insensitive to the loss of Stu2. Would mammalian neurons be similarly less sensitive to the loss of XMAP215?

3. The authors claim at several points that the V409 mutants delay or interfere with neurite retraction. This claim is not supported by their data. For example, on line 190-191, the authors state just two ways in which the V409 mutations might affect neurite growth: "(1) insufficient neurite extension, or (2) insufficient neurite retraction." There are more possibilities, such as ectopic neurite growth, which is different from neurite retraction (i.e., there is difference between a neurite that a wild-type neuron might extend and retract versus an additional, ectopic neurite that a wild-type neuron would not normally extend). The authors consider more than just these two possibilities.

The main support for the idea that neurite retraction is disrupted seems to be the cold treatment assay shown in Figure 2C. This seems an odd assay to test whether the V409 mutations affect neurite retraction. Why not just analyze neurite growth and retraction over time, without any cold treatment? If neurons expressing V409 mutant tubulin have abnormal neurite retraction, this should be determined by visualizing the growth and retraction of a neurite over time using a marker such as GFP, not β-tubulin. A time-lapse imaging experiment would provide convincing data on whether neurite retraction is affected.

Cold is typically used to destabilize microtubules, and, indeed, the label shown in Figure 2C is for β-tubulin. Instead of neurite retraction, it seems that a conclusion about microtubule stability could be drawn from these experiments -? But these data are difficult to interpret because they are not quantified. And the authors should state in the main text when the neurons are shifted to 4°C (the control neuron at 0h appears to be Stage 2).

Most importantly, however they are used, these data in Figure 2C need to be quantified.

4. Figure 2B: The authors quantify branches (or neurites), but are these axons? Dendrites? Are the secondary and tertiary branches extending from axons or dendrites? The authors need to clarify whether axons or dendrites are being quantified.

5. Lines 450-452: The authors claim that their "data support a hypothesis that TUBA1A-V409I/A mutants disrupt and/or delay proper morphology transitions." This conclusion is not supported by their data. The authors do not analyze neuronal morphology in the mouse cortices expressing V409 mutant tubulin, and the authors found the V409 mutant tubulin does not affect the morphological transition of cultured cortical neurons from Stage 2 to Stage 3.

6. Figure 3: The authors quantify tyrosination and acetylation along a presumably significantly large stretch of axons and dendrites (the actual length should be reported in the methods). Given that these PTMs vary in intensity along the neurite length (e.g. tyrosinated tubulin is reportedly highest near the tips of axons or growth cones), could this approach be missing a potential phenotype?

7. Lines 527-531: It would be helpful to know how specifically how (the loss of) XMAP215 affects developing neurons. The current description is quite vague.

*Reviewer #2 (Recommendations for the authors):*

The authors should revise the discussion and introduce structural models to present the defects and their impact on the structural organization in the early part of the paper (Figure 1 or its supplements) The structural presentation can also be included in figure discussing the structural models for TOG proteins figure 4 and some discussion or potential experiments measuring the impact on TOG proteins concentrations on the tubulin concentration in cells.

*Reviewer #3 (Recommendations for the authors):*

– In figure 2B, data points are represented as single cells (at least 27 cells per condition). What is the number of replicates for this. How were the statistics performed? Here average values of independent replicated should be displayed and statistical tests should be performed on these numbers.

– Throughout the manuscript, the precise Ns should be displayed (number of cells per condition, number of replicates), and the statistical tests should be presented in the legends, not in the methods.

– 2C and S1C are the same. The point of this was not clear.

– Line 80: "been studied, and more continued to be identified in the clinic."

---

## [Author Response]

Essential revisions:1) To determine if the results from their yeast model that the V409 mutations have a strong effect on microtubule polymerization are consistent in neurons, the authors should assay directly for an effect on microtubule polymerization in neurons using a +TIP marker, such as EB3-GFP.

We agree with the reviewers that it would strengthen the findings in yeast to assay the effect of the V409 mutants on microtubule polymerization in neurons. To address this, we transfected primary cortical neuron cultures with a pCIG2 plasmid that co-expresses TUBA1A (either WT or mutant) with GFP-MACF43, a plus-end binding protein that binds to EB1 and tracks growing microtubule plus ends (PMID: 19632184).

We find that the speeds of GFP-MACF43 comets are faster in V409I, and more significantly V409A expressing neurons, compared to WT controls. This indicates that V409I and V409A mutants lead to increased microtubule polymerization rates as compared to WT. This data is now presented in Figure 3E-H. These results are in agreement with our yeast data.

2) As suggested by Reviewer 1, please define the neurites you are analyzing. Even if you cannot assign them as axons or dendrites, please define them as primary, secondary, tertiary, etc., as is common across the neurobiology field.

We thank the reviewer for pointing this out, and we have now clarified this in Figure 2B to read as the number of processes/cell. The longest neurite was considered to be the axon as these were stage 3 neurons, however we did not stain for an axonal marker, and therefore we are unable to definitively call it the axon. Additionally, the secondary and tertiary branches quantified in Figure 2B and Figure 2—figure supplement 1B extended from both axons and dendrites.

3) Both Reviewer 1 and 3 expressed concerns about the analysis and conclusions of the neurite retraction assays. Please address both comments, but especially please see Reviewer 1's helpful third comment about convincing ways to analyze neurite retraction.

We thank the reviewers for this suggestion, and we agree that this would be a valuable

experiment. To test this, we have now imaged DIV2 neurons expressing either WT or V409 mutant tubulin and cytoplasmic GFP every 30 seconds over a 15-minute time course in a stagetop incubator at 37°C and 5% CO2. We then measured the length of neurites and branches from the base of the process to the distal tip at each time point. We find that the growth and retraction rates are similar between WT and V409 mutants, however the V409 mutants have more processes that remain in a paused state and fewer that are retracting. These data are shown in Figure 2C-E and Table 1.

4) Both Reviewer 1 and 3 noted that conclusions about morphology transitions are not supported by the data presented in the paper. Please either address this experimentally as suggested by the reviewers, or remove this conclusion. The authors should also address the other possibilities such as a delay in neuronal migration, as suggested by Reviewer 3.

We appreciate the suggestion by the reviewers to analyze the cells in the WT and V409 mutant in utero electroporation images. To address this, we used a Stellaris 5 confocal microscope with 5 HydS tunable detectors to obtain high-resolution images of cells in the ventricular/subventricular and intermediate zones (as defined in Figure 1D). We find that particularly V409A expressing cells have highly complex cell morphologies, as compared to the bipolar morphology of WT cells. These images are now included as a new panel F in Figure 1.

It is interesting that V409I expressing cells also appear to have a more bipolar morphology as compared to V409A, but still have many more cells in these regions as compared to WT. This suggests to us that there may be a delay in neuronal migration, as suggested by Reviewer 3. We have addressed this point in the discussion, page 16 lines 531-536.

While we agree that delayed neuronal migration is an intriguing potential explanation for both V409 mutants, comparing rates of neuronal migration would best be addressed by a combination of a time course of in utero electroporation experiments and ex vivo assays of living cells (i.e. slice cultures). We feel this is beyond the scope of the current manuscript and would be better suited as a central focus of future studies.

5) During the discussion, all three reviewers raised concerns about the lack of data and over-interpretation for how the mutants relate to the conformation of the tubulin dimer. At the very least, the authors should provide a structure of the tubulin dimer with the residues highlighted. This can easily be done in pymol or a similar program. Perhaps once the authors include this data, they may modify their conclusions on whether these mutations could feasibly affect dimer conformation, or if, based on the residue locations, there are other models that the authors may want to discuss. Reviewer 2 has the most thorough review on this point.

We understand the reviewers’ concerns about the over-interpretation of how these mutants could alter the conformation of the tubulin dimer as we did not directly test this idea. As suggested, we have included a PyMOL structure of the tubulin dimer as a new panel A in Figure 5 with the V409 residue of interest highlighted.

Additionally, we have softened the language in the discussion as we did not directly test this model. However, we feel it is still important to discuss the possibility of these mutants altering the conformation of the heterodimer, particularly since we see increased microtubule polymerization rates in the V410A in vitro microtubule dynamics assays. These results clearly indicate that the V410A mutant has an intrinsic effect on microtubule dynamics. Additionally, we see reduced binding between both mutants and TOG domains, which preferentially bind to the kinked conformation of tubulin. Nevertheless, we agree that without directly testing these predictions, we cannot make strong conclusions about tubulin. We believe that our revised

discussion (p18-19, lines 603-619) strikes the appropriate balance of presenting altered tubulin conformation as one of several possible models to consider.

6) It was agreed in the reviewer discussion that the authors should discuss potential effects on the total amount of tubulin available for polymerization if TOG-tubulin binding is inactivated. This could be the reason for the enhancement in microtubule polymerization rates observed with the mutants and would be consistent with most of the data.

We thank the reviewers for this suggestion and have added it to the discussion (p18, lines 604-611) as well as a potential explanation in our model Figure 7.

While we think this is an interesting idea and could certainly be playing a role in vivo, it is interesting to note that V410A drives faster microtubule polymerization rates in vitro, in the absence of any extrinsic regulators and where we can compare experiments with a similar range of tubulin concentrations. The results of our in vitro experiments do not support the model where an increased concentration of mutant tubulin is sufficient to explain increased microtubule polymerization rates.

However, we have also added more discussion regarding the various activities of

XMAP215/Stu2 (p19, lines 628-631). We think this is an interesting and somewhat unexplored area of neuronal development and are very interested in further understanding XMAP215’sroles in cells.

7) Finally, please provide precise n for the number of cells per condition and the number of replicates. In addition, please present the statistical tests in the legends, as is outlined in the eLife guide to authors.

We have now provided the precise number of cells per condition, the number of replicates, and statistical tests in all figure legends.

Reviewer #1 (Recommendations for the authors):1. Given the findings from their yeast model that the V409 mutations have a strong effect on microtubule polymerization, the authors should assay directly for an effect on microtubule polymerization in neurons (e.g. using EB3-GFP).

Please see our response to Essential Revisions #1.

2. The authors find that yeast V410 mutants are insensitive to the loss of Stu2. Would mammalian neurons be similarly less sensitive to the loss of XMAP215?

This is an interesting suggestion. We agree with the reviewer’s speculation that microtubule dynamics in neurons expressing V409 mutant TUBA1A should be less sensitive to the loss of XMAP215. However, we would point out that even though V410 mutant yeast cells retain microtubules during Stu2 depletion, Stu2 is still essential in these cells. It may be that Stu2 retains some level of activity toward V410 mutant tubulin, which may be exquisitely important in some contexts; e.g., at the yeast kinetochore.

3. The authors claim at several points that the V409 mutants delay or interfere with neurite retraction. This claim is not supported by their data. For example, on line 190-191, the authors state just two ways in which the V409 mutations might affect neurite growth: "(1) insufficient neurite extension, or (2) insufficient neurite retraction." There are more possibilities, such as ectopic neurite growth, which is different from neurite retraction (i.e., there is difference between a neurite that a wild-type neuron might extend and retract versus an additional, ectopic neurite that a wild-type neuron would not normally extend). The authors consider more than just these two possibilities.The main support for the idea that neurite retraction is disrupted seems to be the cold treatment assay shown in Figure 2C. This seems an odd assay to test whether the V409 mutations affect neurite retraction. Why not just analyze neurite growth and retraction over time, without any cold treatment? If neurons expressing V409 mutant tubulin have abnormal neurite retraction, this should be determined by visualizing the growth and retraction of a neurite over time using a marker such as GFP, not β-tubulin. A time-lapse imaging experiment would provide convincing data on whether neurite retraction is affected.Cold is typically used to destabilize microtubules, and, indeed, the label shown in Figure 2C is for β-tubulin. Instead of neurite retraction, it seems that a conclusion about microtubule stability could be drawn from these experiments -? But these data are difficult to interpret because they are not quantified. And the authors should state in the main text when the neurons are shifted to 4°C (the control neuron at 0h appears to be Stage 2).Most importantly, however they are used, these data in Figure 2C need to be quantified.

We agree with the reviewer that there are more than two possibilities of how the V409 mutants may affect neurite growth and agree that we did not thoroughly explore alternative possibilities in the original manuscript. We have now updated this section in the results on page 7, lines 218-221 to consider other possibilities. Additionally, we now address these possibilities by conducting new, time-lapse imaging of neurite dynamics in neurons cultured in standard conditions, as suggested by the reviewer. These results are now included in Figure 2C-E and Figure 2—figure supplement 1C. As explained further in our response to Essential Revisions point #3, above, the results of these new experiments reveal that neurites in cells expressing V409 mutants spend more time in a paused state and less time in a retraction state. We thank the reviewer for this suggestion.

We also agree with the reviewer that the cold shock assay is a better test of microtubule

stability. We have therefore rewritten our interpretation of these results on pages 8-9 lines 261-282, and included new quantification of the data in Figure 2F-H and Figure 2—figure supplement 1E-F.

4. Figure 2B: The authors quantify branches (or neurites), but are these axons? Dendrites? Are the secondary and tertiary branches extending from axons or dendrites? The authors need to clarify whether axons or dendrites are being quantified.

Please see our response to Essential Revisions point #2. In the revised manuscript we have now clarified this in Figure 2B to read as the number of processes/cell.

5. Lines 450-452: The authors claim that their "data support a hypothesis that TUBA1A-V409I/A mutants disrupt and/or delay proper morphology transitions." This conclusion is not supported by their data. The authors do not analyze neuronal morphology in the mouse cortices expressing V409 mutant tubulin, and the authors found the V409 mutant tubulin does not affect the morphological transition of cultured cortical neurons from Stage 2 to Stage 3.

Please see our response to Essential Revisions point #4 and the new Figure 1F panel in the revised manuscript. We believe that studying the morphology transitions in the in utero electroporations is more valuable than in in vitro cultures because cells in vivo must respond to more developmental cues at the right place and time to make these transitions.

6. Figure 3: The authors quantify tyrosination and acetylation along a presumably significantly large stretch of axons and dendrites (the actual length should be reported in the methods). Given that these PTMs vary in intensity along the neurite length (e.g. tyrosinated tubulin is reportedly highest near the tips of axons or growth cones), could this approach be missing a potential phenotype?

We thank the reviewer for this suggestion. We have now adjusted our analysis and binned the PTM quantification in two 10μm segments at the distal end of the axon and two 5μm segments at the distal end of one dendrite. These data are now found in Figure 3A-D and Figure 3-figure supplement 1A-F. Our new analysis shows that, compared to WT, there is an increase in microtubule acetylation in V409I/A axons, and particularly high levels at the very distal tip of V409A axons. However, we find that there is no significant increase in microtubule acetylation at the distal ends of the dendrite in the V409 mutants as compared to WT. These details have also been updated in the methods section.

7. Lines 527-531: It would be helpful to know how specifically how (the loss of) XMAP215 affects developing neurons. The current description is quite vague.

We thank the reviewer for this input and agree that this would also strengthen our discussion section and have expanded on this in Essential Revisions point #6 and have now made updates on p19, lines 628-631.

Reviewer #2 (Recommendations for the authors):The authors should revise the discussion and introduce structural models to present the defects and their impact on the structural organization in the early part of the paper (Figure 1 or its supplements) The structural presentation can also be included in figure discussing the structural models for TOG proteins figure 4 and some discussion or potential experiments measuring the impact on TOG proteins concentrations on the tubulin concentration in cells.

We thank the reviewer for this input and careful review of the tubulin conformation portion of this manuscript. We have now added a PyMOL structure of where TUBA1A-V409 resides on the heterodimer using PDB: 3J6F in Figure 1—figure supplement 1A and describe how this residue may interact within the dimer and on the external side of the microtubule (page 5 lines 147-149).

Additionally, we have added another PyMOL structured modified from PDB: 4FFB in Figure 5A where yeast α/β-tubulin is complexed with the TOG1 domain of Stu2 to highlight how residue V410 may interact with Stu2.

We agree with the reviewer that it would be interesting to measure the impact of TOG

concentrations on the free tubulin concentration available in cells. This is an experiment we have thought about previously, and we predict that overexpressing TOG domains in cells would bind up and saturate the free tubulin pool, and ultimately hinder microtubule polymerization. Because we see decreased binding affinity between TOG domains and the V410I/A mutants, we would predict that mutant microtubules would still assemble in the presence of overexpressed TOG domains.

Reviewer #3 (Recommendations for the authors):– In figure 2B, data points are represented as single cells (at least 27 cells per condition). What is the number of replicates for this. How were the statistics performed? Here average values of independent replicated should be displayed and statistical tests should be performed on these numbers.

We have now provided the precise number of cells per condition, the number of replicates, and statistical tests in all figure legends. We have also made these plots into superplots to display the average values of independent replicates.

– Throughout the manuscript, the precise Ns should be displayed (number of cells per condition, number of replicates), and the statistical tests should be presented in the legends, not in the methods.

Please see response above and Essential Revisions #7.

– 2C and S1C are the same. The point of this was not clear.

Thank you to the reviewer for pointing this out. Our intention was to show that the images shown in the main figure were expressing either WT or V409 mutant tubulin, as indicated by the cytoplasmic GFP. We have now pointed this out more clearly on page 8 lines 270-271.

– Line 80: "been studied, and more continued to be identified in the clinic."

We have now made this adjustment and the line now reads, “Only a small fraction of *TUBA1A* mutants have been studied, and more continue to be identified in the clinic.”